



# Comparison of Holocene temperature reconstructions based on GISP2 multiple-gas-isotope measurements

Michael Döring[1,2*] and Markus Christian Leuenberger[1,2]

[1]Climate and Environmental Physics, University of Bern, Switzerland
[2]Oeschger Centre for Climate Change Research (OCCR), Bern, Switzerland

*Correspondence to: Michael Döring (michael.doering@climate.unibe.ch)

Keywords: temperature reconstruction, ice core, nitrogen isotope, argon isotope, inverse-model, firn-model, accumulation-rate

**Abstract:** Nitrogen and argon stable-isotope data extracted from ancient air in ice cores provides the possibility
to reconstruct Greenland past temperatures when inverting firn-densification and heat-diffusion models (firn-models) to fit the gas-isotope data ($\delta^{15}N$, $\delta^{40}Ar$, $\delta^{15}N_{excess}$). This study uses the Döring and Leuenberger (2018) fitting-algorithm coupled on two state of the art firn-models to fit multiple Holocene gas-isotope data measured on the GISP2 ice core. We present for the first time the resulting temperature estimates when fitting $\delta^{15}N$, $\delta^{40}Ar$ and $\delta^{15}N_{excess}$ as single targets with misfits generally in the low permeg level. Whereas the comparison between
the reconstructions using $\delta^{15}N$ and $\delta^{40}Ar$ shows a high agreement, the use of $\delta^{15}N_{excess}$ for reconstructing temperature is problematic, due to higher statistical and systematic data uncertainty influencing especially multi-decadal to multi-centennial signals, and results in an unrealistic temperature estimate that differs significantly from the two other reconstructions. We find evidence for systematic too high $\delta^{40}Ar$ data in the early- and late-Holocene potentially caused by post coring gas-loss or an insufficient correction of this mechanism. Next, we compare the
performance of the Goujon et al. (2003) firn-model and the Schwander et al. (1997) firn-model for Holocene temperature reconstructions. Besides small differences of the reconstructed temperature anomalies – potentially caused by slightly different implementation of firn physics and parameters in the two models – the reconstructed temperature anomalies are highly comparable. We were able to quantify the contribution of the firn-model difference to the uncertainty budget of our reconstruction. Furthermore, the fractions of uncertainty on the
reconstructed temperatures, arising from the non-perfect reproducibility of the fitting algorithm and from the remaining final misfits (low permeg level), were quantified. Together with the published measurement uncertainty of the gas-isotope data and the analysis of the impact of accumulation-rate uncertainty on the reconstruction, we were able to calculate the mean uncertainty ($2\sigma$) for the nitrogen and the argon based temperature estimates with $2\sigma_T = 0.80…0.88$ K for $T(\delta^{15}N)$, and $2\sigma_T = 0.87…1.81$ K for $T(\delta^{40}Ar)$, respectively. Finally, we compare our
reconstructed temperatures to two recent reconstructions based on the same gas-isotope data as used here, but following different reconstruction strategies: first the study of Buizert et al. (2018), which uses a combination of $\delta^{18}O_{ice}$-calibration and $\delta^{15}N$-fitting, and second the study of Kobashi et al. (2017), where $\delta^{15}N_{excess}$ was fitted in order to conduct the temperature reconstruction. We find generally higher agreement between our $T(\delta^{15}N)$ estimate and the Buizert et al. (2018) temperature – in terms of variability and correlation in three investigated periodic-
time bands (multi-decadal, multi-centennial and multi-millennial) – as if our $T(\delta^{15}N)$ reconstruction is compared to the Kobashi et al. (2017) temperature. However, all three reconstruction strategies lead to distinct temperature realizations.





## 1 Introduction

The use of nitrogen ($\delta^{15}$N) and argon ($\delta^{40}$Ar) stable-isotope variations in air extracted from ice cores is a relatively new tool for reconstructing past temperature (e.g. Huber et al., 2006b; Kindler et al., 2014; Kobashi et al., 2011; Landais et al., 2006; Orsi et al., 2014; Severinghaus et al., 1998, 2001). This method uses the stability of isotopic compositions of nitrogen and argon in the atmosphere at orbital timescales, as well as the fact that changes are only driven by processes in polar firn (Leuenberger et al., 1999; Mariotti, 1983; Severinghaus et al., 1998) and provides an alternative to the classical calibration of the stable oxygen ($\delta^{18}$O) and hydrogen isotopes extracted from the ice-core-water samples (Gierz et al., 2017; Johnsen et al., 2001; Steen-Larsen et al., 2011; Stuiver et al., 1995). The isotopic composition of the water samples provides a rather robust proxy for reconstructing paleo-temperatures for times where large temperature variations occur (Gierz et al., 2017). In the Holocene, where temperature variations are comparatively small, changes in seasonal distribution of precipitation as well as of evaporation conditions at the source region may dominate water-isotope-data variations (Huber et al., 2006b; Kindler et al., 2014; Werner et al., 2001). Recent studies (Buizert et al., 2018; Kobashi et al., 2017) used the nitrogen and argon isotopes of the GISP2 ice core (Greenland Ice Sheet Project Two, Meese et al., 1994; Rasmussen et al., 2008; Seierstad et al., 2014) to reconstruct Holocene temperature variations for Greenland summit following different reconstruction strategies. Kobashi et al. (2017) use the second-order parameter $\delta^{15}$N$_{excess}$ ($\equiv \delta^{15}$N - $\delta^{40}$Ar/4) together with the firn-densification and heat-diffusion model from Goujon et al. (2003) to obtain a Holocene temperature estimate. Buizert et al. (2018) reconstructed summit temperatures by calibrating the GISP2 $\delta^{18}$O data by forcing the temperature to reproduce the general trend in $\delta^{15}$N using a dynamical firn-model. Both methods lead to different temperature estimates. In Buizert et al. (2018) an overall uncertainty of 1.5 K was stated for the reconstructed temperature. Kobashi et al. (2017) estimated the uncertainty of the temperature reconstruction by examining the variance of temperature realizations when shifting the $\delta^{15}$N$_{excess}$ data in the range of analytical uncertainty before using his fitting approach. This approach results in an averaged uncertainty of 1.21 K (1$\sigma$). Both approaches have in our view shortcomings, $\delta^{15}$N$_{excess}$ loses information about the Lock-in-depth (LID) and scaling to $\delta^{18}$O$_{ice}$ does not consider side-effects of water isotopes to seasonal distribution of precipitation. Döring and Leuenberger (2018) showed an automated approach enabling fitting gas-isotope data with an outstanding accuracy with mismatches generally below the analytical uncertainty of the isotope measurements. It was shown on synthetic data experiments that in the case of perfectly known accumulation-rate data and neglecting noise, the remaining mismatches would lead to a temperature uncertainty (2$\sigma$) below 0.3 K for a single measurement and Holocene-like conditions. This study focuses on the challenges of temperature reconstructions using gas-isotope fitting for real Holocene data. We will discuss different aspects which are in our view integral for the evaluation of the correctness of the reconstructed temperature estimates. First, we will discuss the gas-isotope data in the context of measurement uncertainty and focus on the suitability of the different isotopic quantities for reconstructing robust temperature estimates. Next, we discuss the reproducibility and the contribution of the final misfits to the uncertainty budget using our fitting approach. In addition, we will show the influence of different accumulation-rate estimates on the temperature reconstruction. Finally, we compare the temperature solutions obtained by fitting $\delta^{15}$N, $\delta^{40}$Ar and $\delta^{15}$N$_{excess}$ to each other and place them in a context to the estimates of Kobashi et al. (2017) and Buizert et al. (2018). For our reconstruction we used two different firn-models, the models of Schwander et al. (1997) and of Goujon et al. (2003). We will compare our results using both and we will provide an overall uncertainty of our method for the most robust estimate using all available information.





## 2 Data and method

### 2.1 Firn-models and inversion algorithm

The observed gas-isotope data mainly relies on firn densification processes combined with gas and heat diffusion (Severinghaus et al., 1998). So the use of firn-densification and heat-diffusion models (from now on referred to as

firn-model) describing the physics of densification and heat and gas transport are necessary for inverting measured gas-isotope data to surface temperatures. In this study we use two classic firn-models. The first model was developed by Schwander et al. (1997) and used for the temperature reconstructions by Huber et al. (2006b) and Kindler et al. (2014). The second one, the model from Goujon et al. (2003) (adapted to the GISP2 site for this study), was used i.e. in the studies by Guillevic et al. (2013), Kobashi et al. (2015), and Kobashi et al. (2017). As

a comparison of temperature reconstructions for Holocene conditions using both models is done for the first time in this study, we will prove the comparability of the solutions gained by using both models.

The conversion of gas-isotope data to surface temperature estimates using a firn-model is an inverse problem. The firn-model acts as a non-linear transfer function, combining temperatures and accumulation-rates with the gas-isotope data. To solve this problem we use the automated fitting algorithm developed by Döring and Leuenberger

(2018). The used algorithm allows the fitting of the gas-isotope data with misfits in the low permeg level, mainly below the analytic measurement uncertainties. For modelling $\delta^{40}Ar$ and $\delta^{15}N_{excess}$ data, we use, in addition to the details presented in Döring and Leuenberger (2018), the thermal-diffusion constant $\alpha_{T,Ar}$ and thermal-diffusion sensitivity $\Omega_{Ar}$ that have been empirically derived by Grachev and Severinghaus (2003):

$$\alpha_{T,Ar}(t) = \left(26.08 - \frac{3952}{\bar{T}(t)}\right) \cdot 10^{-3} \tag{1}$$

$$\Omega_{Ar}(t) = \frac{\alpha_{T,Ar} \cdot 10^3}{\bar{T}(t)} = \frac{26.08}{\bar{T}(t)} - \frac{3952}{\bar{T}(t)^2} \tag{2}$$

$\bar{T}(t)$ is the mean firn temperature (Leuenberger et al., 1999).

### 2.2 Timescale, and necessary data

*Timescale*

For the entire study, the GICC05 chronology is used (Rasmussen et al., 2014; Seierstad et al., 2014). Following Döring and Leuenberger (2018), the temperature input is split into two parts in time. The first part ranges from 10.5 kyr to 35 kyr b2k ("spin-up section"), whereas the second part ranges from 0.02 kyr to 11.5 kyr b2k

("reconstruction window") for which we allow the fitting algorithm to change the temperature. The accumulation-rate as well as the surface temperature of the spin-up section remain unchanged during the reconstruction procedure.

*Accumulation-rate input data*

In addition to temperature, accurate accumulation-rate data are needed to drive the firn-models. As in Döring and

Leuenberger (2018), we use as accumulation-rate input the original accumulation-rates (acc) for the GISP2 site, as reconstructed in Cuffey and Clow (1997), but adapted to the GICC05 chronology. We use for the reconstruction all three given accumulation-rate datasets to analyse the impact of differences between the three accumulation estimates on the reconstructed temperatures. To obtain the accumulation rate estimates three different scenarios of





ice sheet margin retreat were used (50 km, 100 km or 200 km scenario) as boundary conditions for an ice flow model (see Cuffey and Clow (1997) for details; availability of the accumulation rates using the 100 km scenario on GICC05 timescale: //doi.pangaea.de/10.1594/PANGAEA.888997).

*Target data and their suitability due to analytical uncertainty*

The GISP2 gas-isotope data measured by Kobashi et al. (2008b) will be used for the Holocene temperature reconstruction as targets for our fitting algorithm (Döring and Leuenberger, 2018). Figure 1 shows the different gas-isotope targets ($\delta^{15}N$, $\delta^{40}Ar$, $\delta^{15}N_{excess}$) set on the GICC05 chronology. In tab. S1 we list measurement uncertainties for the target data given in Kobashi et al. (2008b). As different uncertainties were given in Kobashi et al. (2008b) for different measurement campaigns, we consequently use the minimal and maximal uncertainties

given in Kobashi et al. (2008b) for a signal-to-noise analysis of these data for two cases, signals with periodicities $T < 500$ yr ("high frequency") and $T > 500$ yr ("low frequency"), respectively. Here we focus especially on the faster signal ($T < 500$ yr) in order to analyse the suitability in relation to the analytical uncertainty of the different isotope targets ($\delta^{15}N$, $\delta^{40}Ar$, $\delta^{15}N_{excess}$) for reconstructing multi-decadal to multi-centennial temperature variability. We conducted the analysis in the following way: First, we detrended the measured gas-isotope target data by

subtracting the respective low-pass-filtered signals using a cut-off-period of 500 yr (fig. S1a-c). In a next step, we identified the local maxima and minima of the high-frequency isotope data (using the Matlab® "findpeaks" algorithm). We defined the high-frequency signals as the difference between successive local maxima and minima (fig. S1d-f) and compared the high-frequency signals to the signal uncertainties calculated from the published measurement uncertainties (Kobashi et al., 2008b). As a signal is defined by at least two points, the signal

uncertainties are calculated using Gaussian-error-propagation. We use for our calculations the minimum (red dotted line, fig. S1) and maximum (blue dotted line, fig. S1) uncertainties (tab. S1). The analysis of the high-frequency signals shows that for the minimum uncertainties 78% of the $\delta^{15}N$ high-frequency signals have amplitudes higher than the uncertainty level (70% for the maximum uncertainty), 74% (or 36%) for $\delta^{40}Ar$, and only 52% (or 17%) for $\delta^{15}N_{excess}$, respectively. Assuming that the "true" uncertainty is in-between the given

maximum and minimum uncertainties and keeping in mind that the listed measurement uncertainties are 1σ uncertainty, we argue that only $\delta^{15}N$ is suitable as a robust reconstruction target in the high-frequency case. The histograms in fig. 1 show a detailed listing of the signal-to-noise ratios (SNRs) for all isotope species and for the minimum and maximum uncertainties, respectively. Here the signals are grouped between integer SNR values. It is clearly visible that for $\delta^{15}N$ most signals have an SNR between one and two or even higher values for the

minimum as well as the maximum uncertainty. In contrast, for $\delta^{40}Ar$ and especially for $\delta^{15}N_{excess}$ the dominant fraction of signals has SNR values lower than the uncertainty values (SNR < 1), which makes it challenging to extract a robust temperature estimate for multi-decadal to multi-centennial signals from these targets.

For the longer-term isotope trends ($T > 500$ yr) it is more challenging to extract a comparable result (fig. S1g-i). Here we divided the analytical uncertainty by a factor of about 5.3 to account for the smoothing (mean data

resolution = 17.8yr, $(500 \text{ yr}/17.8 \text{ yr})^{1/2} = 5.3$). The comparison of the minimum or maximum measurement uncertainties (red or blue error bars) indicates that all three gas-isotope quantities are suitable for reconstructing long-term temperature trends. This is particularly correct when only measurement uncertainty is considered, as these uncertainties are in most of the cases lower than the amplitudes of the investigated features. However, $\delta^{15}N$ is also the most suitable target for reconstructing long-term temperature trends due to its relatively small

uncertainty compared to $\delta^{40}Ar$ and $\delta^{15}N_{excess}$.



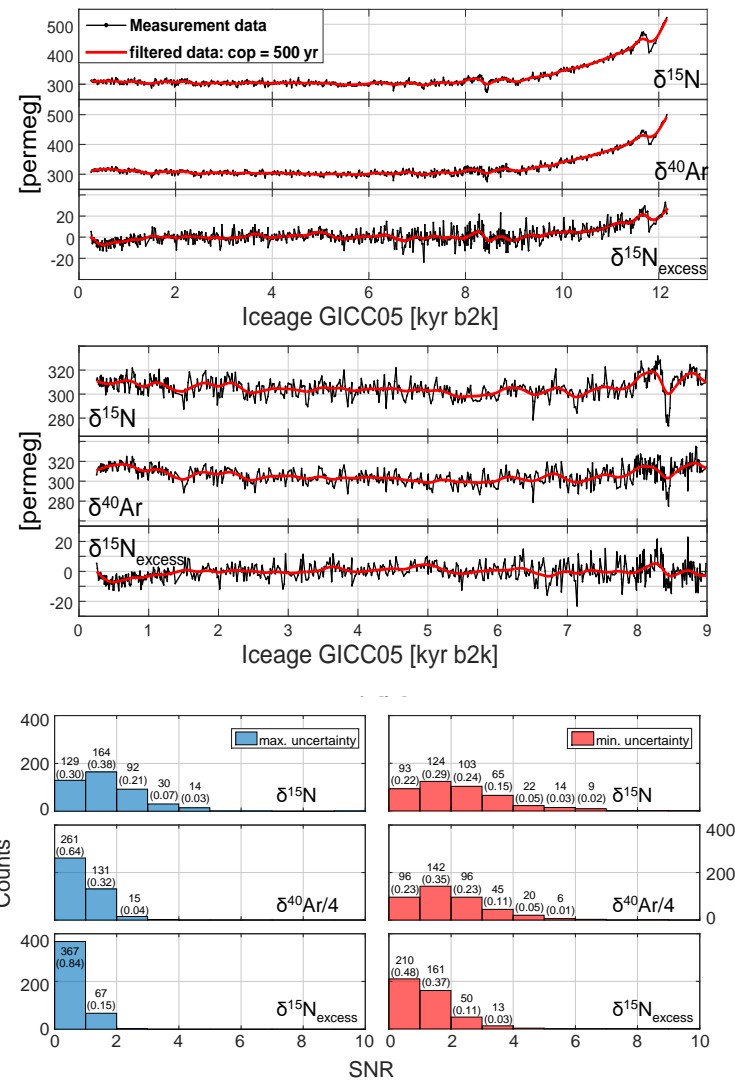

**Figure 1: Gas-isotope target data on GICC05 time scale (Kobashi et al., 2008b): Upper three plots: full δ15N, δ40Ar/4 and δ15Nexcess time-series; Middle three plots: zoom in for the recent 9 kyr time window of the same quantities. Histograms: Signal-to-noise-ratios (SNR) of the high-frequency signals (T < 500 yr) for all reconstruction targets and for the maximum (left hand side) as well as the minimum (right hand side) signal uncertainties (see text). Values in the plots indicate the absolute (relative) number of signals with SNR in between the respective limits.**

*Problems with δ15Nexcess due to gas-loss fractionation:*

It must be stressed that using δ15Nexcess only (as single target) for the reconstruction involves the danger of incorporating large drifts in the temperature solution if systematic offsets in δ15Nexcess are present, since the calculated temperature is solely dependent on the temporal integration of firn-temperature gradients directly calculated from δ15Nexcess. This can alter centennial or even millennial scale temperature variability. It is known that smaller molecules such as Ar suffer from kinetic fractionation due to gas-loss either during bubble close-off or core retrieval and storage which can lead to an enrichment of δ40Ar and thus to smaller δ15Nexcess (Huber et al.,





2006a; Kobashi et al., 2008b, 2010, 2011; Severinghaus et al., 2003; Severinghaus and Battle, 2006). This made a correction of the GISP2 $\delta^{40}Ar$ data mandatory, which was done based on firn-modelling (Kobashi et al., 2008a) and on $\delta Ar/N_2$ (a tracer for potential gas-loss; Kobashi et al., 2010, 2017), leading to smaller $\delta^{40}Ar$ values compared to the uncorrected $\delta^{40}Ar$ and thus to higher $\delta^{15}N_{excess}$. As we will show in sects. 4.1.2 and 4.3, the executed

correction is non-sufficient, especially for the late- and early-Holocene data, which implies that the gas-loss induced fractionation in $\delta^{40}Ar$ is not constant over the Holocene part of the GISP2 ice core. A better correction approach would be the use of additional isotope quantities ($\delta^{86}Kr$, $\delta^{136}Xe$) measured together with $\delta^{40}Ar$ on the same samples as proposed by Baggenstos (2015). Additionally, the elimination of the gravitational signal using $\delta^{15}N_{excess}$ as single target leads to a loss of information (firn column hight) and to a less constraint temperature

solution with reduced reproducibility and this complicates the firn-model inversion (sect. 3.1.1).

The linear dependency (slope) between $\delta^{15}N$ and $\delta^{40}Ar/4$ (tab. 1) is calculated using geometric-mean-regression (Leng et al., 2007) in order to investigate the contribution of different processes altering the isotope data. Furthermore, the slope is calculated in three periodic-time bands (multi-decadal, multi-centennial, and multi-millennial) and for the raw data. As the variability of the isotope data is created in the firn due to gravitational

settling (mass dependent fractionation process) and thermal diffusion (dependent on the firn temperature gradient), the slope between $\delta^{15}N(y)$ and $\delta^{40}Ar/4(x)$ should be in the range of one (gravitational settling only) to 1.46 (thermal diffusion only, $\Omega_N/[\Omega_{Ar}/4]$). Also, both processes affecting $\delta^{15}N$ and $\delta^{40}Ar$ are in the same "direction" and in consequence, a generally high correlation is expected. If we compare the results for the different periodic-time bands, it is obvious that the slope calculated for the multi-decadal oscillations cannot be explained neither by

gravitational settling nor by thermal diffusion. For these fast oscillations the gravitational background is not expected to change significantly, and the slope should correspond mainly to the thermal diffusion value of 1.46. In contrast, the calculated slope is $0.89 \pm 0.05$ and the correlation is relatively weak ($r^2 = 0.46$). A slope less than one is pointing to a process which further enriches $\delta^{40}Ar/4$ compared to $\delta^{15}N$, which is the case for a potential gas-loss contribution on $\delta^{40}Ar$ which seems still remaining after the correction of $\delta^{40}Ar$. The weak correlation shows

a decoupling between $\delta^{15}N$ and $\delta^{40}Ar$ for multi-decadal variability, which can be partly attributed to analytical uncertainty. For the longer periodicities we find higher correlation ($r^2 > 0.8$) and slopes in the range of expectation (tab. 1). The decrease of the slope from multi-centennial to multi-millennial variability shows that with longer periodic-time the influence of gravitational fractionation due to changes of firn column height gets more and more important compared to the thermal diffusion signal. This is expected as (i) the firn column reacts slowly and (ii)

the temperature gradient over the firn column vanishes.





| Periodic-time band | $r^2$ | slope |
|---|---|---|
| multi-decadal (50-200 yr) | 0.46 | $0.89 \pm 0.05$ |
| multi-centennial (200 yr-1 kyr) | 0.87 | $1.29 \pm 0.04$ |
| multi-millennial (1 kyr-4 kyr) | 0.87 | $1.14 \pm 0.03$ |
| unfiltered data | 0.96 | $1.14 \pm 0.02$ |

| theoretical values: | |
|---|---|
| thermal diffusion only | 1.46 |
| gravitational settling only | 1.00 |

**Table 1: Slopes and correlation coefficients between $\delta^{15}N(y)$ and $\delta^{40}Ar/4(x)$ derived using geometric-mean-regression. The theoretical value for thermal diffusion was calculated as ratio of the thermal-diffusion-sensitivities as $\Omega_N/[\Omega_{Ar}/4]$.**

*Prior input and model spin-up:*

To avoid the influence of possible memory effects (influence of earlier firn-state conditions on later firn-states), a temperature and accumulation-rate spin-up is needed in order to bring the firn-model to a well-defined starting condition. The surface temperature spin-up was obtained by extending the temperature reconstruction for the GISP2 site from Buizert et al. (2014) (interval 10.05 kyr to 20 kyr b2k) to 35 kyr b2k by calibrating the GISP2 $\delta^{18}O_{ice}$ data (Grootes et al., 1993; Grootes and Stuiver, 1997; Meese et al., 1994; Steig et al., 1994; Stuiver et al., 1995; data availability: Grootes and Stuiver, 1999), using the slopes and intercepts given in Cuffey and Clow (1997). As prior input for the Holocene section, we simply start with constant temperature using the last value of the spin-up section. The prior temperature input and the spin-up temperature were slightly adjusted in order to match the decreasing flank at the oldest part (9.5 kyr to 12.168 kyr b2k) of the gas-isotope data. The adjustment procedure is described in detail in supplement sect. S2.

**3 Results and discussion: gas-isotope fitting**

**3.1 Gas-isotope fitting results and uncertainty due to methodology**

In the following paragraphs we evaluate different factors contributing to the final uncertainty budget of the temperature reconstructions when fitting all different targets ($\delta^{15}N$, $\delta^{40}Ar$ and $\delta^{15}N_{excess}$). These factors are:

(i)  The reproducibility of the resulting solutions over ten fitting runs (sect. 3.1.1). As we use a Monte-Carlo based method, each fitting run follows a unique pathway. In consequence, the obtained final temperature estimates slightly differ between different fitting runs on the very same target.

(ii)  The "goodness" of the fits (sect. 3.1.1). As discussed in Döring and Leuenberger (2018), it is not possible to fit the gas isotope data with an overall misfit of zero. The remaining final mismatches between modelled and measurement data contribute to the uncertainty of the reconstruction.

(iii)  The uncertainty in the used accumulation rate data was incorporated in the temperature reconstruction (sect. 3.1.3), as we used all three different accumulation rate data sets for the GISP2 site provided by Cuffey and Clow (1997).



(iv)    Differences in the reconstructed temperatures resulting from the usage of different firn-models (sect. 4.1.1). This is sketched by analysing the differences between the reconstructed temperatures using two different firn-models. It would be beneficial to incorporate more available models to better quantify the uncertainty due to firn-model differences in later studies.

(v)    The fraction of uncertainty in the reconstructed temperatures due to analytical uncertainty in the used isotope data (sect. 4.2).

### 3.1.1 Reproducibility and final misfits:

In order to analyse the reproducibility of the gas-isotope fitting algorithm coupled to the Schwander et al. (1997) firn-model, ten fitting runs were conducted for each gas-isotope species ($\delta^{15}$N, $\delta^{40}$Ar and $\delta^{15}$N$_{excess}$). From the ten

solutions for each isotope a mean-solution was calculated as the average of the ten temperature solutions. This mean-solution was processed by the firn-model leading to the final modelled isotope solution. The gained data were analysed for the reproducibility between the ten runs (top six plots of fig. 2) to estimate the possible spread in absolute temperature and isotope solutions for each isotope target. In addition, we compare the final (mean) solution with the best-fit solution (out of ten) regarding the remaining mismatches (fig. 3). Table S2 contains

additional information for the reproducibility ("rep") as well as the goodness of the fits ("fit"). It is clearly visible that the spread of the isotope solutions between the ten runs is in the low permeg level for all targets with mean spreads over the whole reconstruction time-window (0.255 kyr to 12.168 kyr iceage b2k) of $1.68 \pm 1.14$ permeg for $\delta^{15}$N, $2.58 \pm 1.77$ permeg for $\delta^{40}$Ar/4, and $1.28 \pm 2.14$ permeg for $\delta^{15}$N$_{excess}$. The low variances between the modelled isotope solutions show the robust performance of our gas-isotope fitting algorithm. Similarly, the spread

between the gained temperatures was analysed (bottom 6 plots of fig. 2). Using $\delta^{15}$N and $\delta^{40}$Ar as single targets lead to temperature solutions varying in a narrow band of $0.17 \pm 0.12$ K for T($\delta^{15}$N) and $0.26 \pm 0.18$ K for T($\delta^{40}$Ar). In contrast to the excellent reproducibility of T($\delta^{15}$N) and T($\delta^{40}$Ar), the fitting of $\delta^{15}$N$_{excess}$ generates a wide spread of possible temperature realizations which contrasts with the robustness of the modelled isotope solutions. Whereas the variance between the isotope solutions is comparable for all three species, the spread of T($\delta^{15}$N$_{excess}$)

is about 10 times ($2.04 \pm 1.90$ K) higher as for T($\delta^{15}$N) or T($\delta^{40}$Ar). This is due to the removal of all information about the gravitational component in the isotope signals when calculating $\delta^{15}$N$_{excess}$, and therefore the information about the height of the firn column is lost, which leads to a significantly extended space of firn-states or absolute temperatures on which the fitting algorithm can work to yield very similar modelled isotope solutions. Besides the spread in absolute temperature, the relative temperature variations (deviation from the trend) gained by fitting

$\delta^{15}$N$_{excess}$ are very much comparable to each other. However, the different densification backgrounds lead to differences in the time evolution of $\Delta$age of about 20 yr to 30 yr, and therefore in asynchronous temperature estimates (fig. S3). Due to this, fitting of $\delta^{15}$N$_{excess}$ as single target makes it challenging to determine the right timing of temperature changes on a multi-decadal scale. Interestingly, fitting of $\delta^{15}$N$_{excess}$ with the Goujon et al. (2003) model leads to a smaller spread of possible temperature estimates compared to the Schwander et al. (1997)

model. The reproducibility between 10 runs (fig. S4) is 2.7 times better when the Goujon model is used. The reason for that difference was not found yet. The implementation of the geothermal heat flux in the Goujon model provides a negative constant fraction to $\Delta$T$_{firn}$ and may lead to this stabilization effect. Figure S3 shows the modelled $\Delta$age and LID data for all fitting targets ($\delta^{15}$N, $\delta^{40}$Ar, $\delta^{15}$N$_{excess}$) and all 10 runs. Whereas the variance between the modelled $\Delta$age and LID is very small for $\delta^{15}$N and $\delta^{40}$Ar, fitting of $\delta^{15}$N$_{excess}$ creates a variety of LID and $\Delta$age

states. While running the gas-isotope fitting algorithm several times on the same target, we notice a boundary



effect for the last 500 yr to 1 kyr to today. Here different temperature solutions emerge, while the rest of the time-series is highly reproducible. To overcome this boundary effect, a stabilization of the solutions for the last 500 yr to today was necessary as described in supplement sect. S4. To solve this issue, additional information is needed, which was added by using the measured borehole temperature profile for the GISP2 site as an additional constraint.

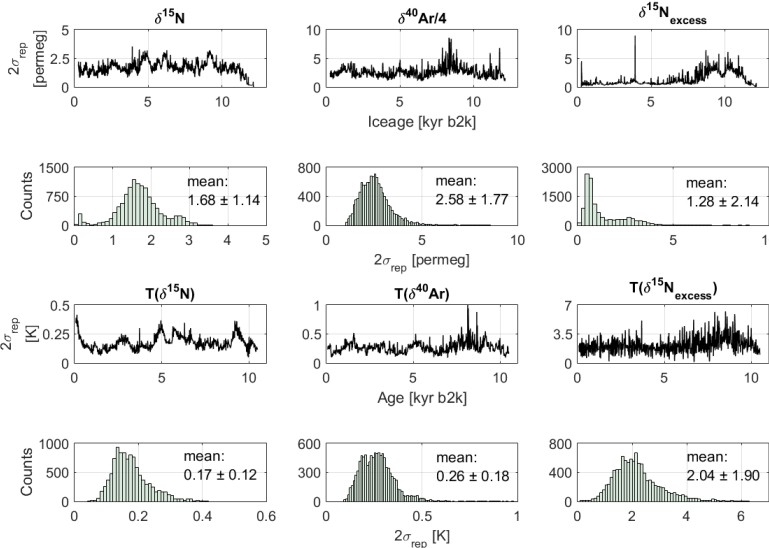

**Figure 2: Reproducibility between 10 runs for each target (first row: $\delta^{15}$N and T($\delta^{15}$N) using the fitting algorithm coupled to the firn-model from Schwander et al. (1997); second row: $\delta^{40}$Ar and T($\delta^{40}$Ar); third row: $\delta^{15}$N$_{excess}$ and T($\delta^{15}$N$_{excess}$)). Top 6 plots: reproducibility for the modelled isotopes per yr. Bottom 6 plots: reproducibility for the reconstructed temperatures per yr. Values are mean ± 2σ. See also table S2.**

10    The evaluation of the mismatches between the measured and modelled time-series gives a constrain on the uncertainty budget of the final temperature. It is obvious that for all targets the mismatches between the measured and modelled isotope data are at least comparable or below the analytic uncertainty (which is 1σ) of the measurement data (tab. S1). Using the average temperatures, we reach final mismatches (2σ) of 3.65 permeg for $\delta^{15}$N, 2.79 permeg for $\delta^{40}$Ar/4, and 5.43 permeg for $\delta^{15}$N$_{excess}$ (fig. 3). Interestingly, averaging the $\delta^{15}$N- and $\delta^{40}$Ar

15    temperature solutions leads to a further decrease of the mismatch compared to the best fit out of the ten runs (4.5 % for $\delta^{15}$N, 18.9 % for $\delta^{40}$Ar). It seems that the averaging of the ten temperature solutions corrects some of the remaining (potentially randomly distributed) mismatches. Obviously, a larger number of runs (> 10) would slightly improve the mean solution. As discussed above, the averaging of the temperature solutions gained from $\delta^{15}$N$_{excess}$-fitting is problematic, due to the wider spread between the temperature solutions and thereof worse constrained

20    Δage. Consequently, the averaging leads to an increase of the mismatches compared to the single fits. The averaging of the single T($\delta^{15}$N$_{excess}$) solutions leads to more than a doubling (factor 2.55) of the mismatches compared to the best-fit solution.

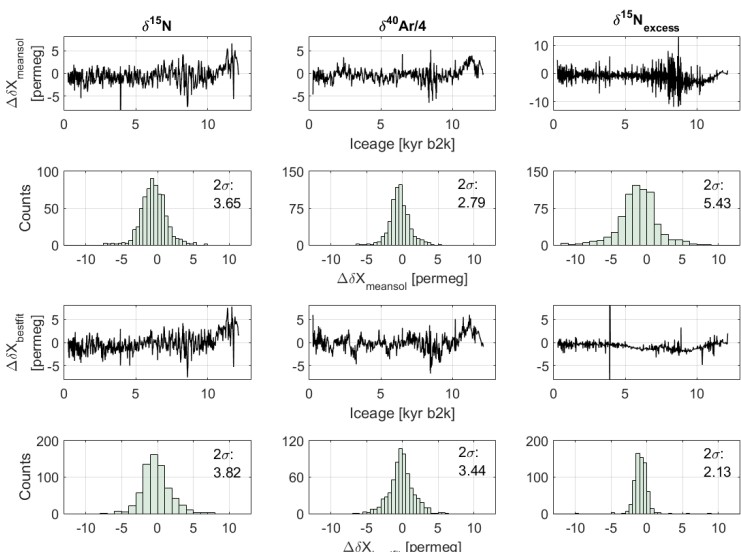

**Figure 3: Top six plots: Mismatches between the modelled and measured isotopes of the final (mean) solution. The numbers in the histograms are the 2σ values of all pointwise mismatches. Bottom 6 plots: Mismatches between the modelled and measured isotopes of the best fit out of ten runs. The numbers in the histograms are the 2σ values of all pointwise mismatches. Interesting is the decrease of the mismatches of the mean solution compared to the best fit for δ15N and δ40Ar. δ15Nexcess shows a reverse behaviour due to the wider spread of "possible" temperature solutions (figure 2 and table S2).**

### 3.1.2 Comparison of misfits among the reconstructions

The histograms on the right-hand-side of fig. 4 show the mean mismatches for all gas-isotope quantities when fitting a single isotope target using the Schwander et al. (1997) (black bars) or the Goujon et al. (2003) (blue bars) firn-model. E.g. precise fitting of $\delta^{15}N$ (top plot, where 96% of the mismatches (2σ) are smaller than 3.7 permeg using Schwander et al. (1997) firn-model) leads to insufficient fits for the other gas-isotope targets with mismatches (2σ values) of 11.3 permeg for $\delta^{40}Ar/4$ and 11.1 permeg for $\delta^{15}N_{excess}$, keeping in mind that the data uncertainty (1σ) of the measured gas isotope data is 3.0 to 4.0 permeg for $\delta^{15}N$, 4.0 to 9.0 permeg for $\delta^{40}Ar/4$ and 5.0 to 9.8 permeg for $\delta^{15}N_{excess}$. In other words, precise fitting of $\delta^{15}N$ does not automatically lead to accurate fits for $\delta^{40}Ar$ and $\delta^{15}N_{excess}$. The same is true for all other single fits and when using both firn-models. It is not possible to find a temperature estimate leading to modelled isotope regimes that provide a sufficient agreement for all isotopic targets together. This finding is pointing to the issues that the GISP2 gas-isotope data suffers from fractionations which are not captured by the used firn-models (e.g. gas-loss).





**Figure 4:** Temperature solutions for all targets with differences between Schwander and Goujon modelling (histogram in the plots), and mean misfits for all species (histogram right hand side).



### 3.1.3 Influence of different accumulation-rate estimates

To investigate the contribution of the uncertainty in the used accumulation-rate data, we use all three available accumulation-rate estimates for the GISP2 site (sect. 2.2) to reconstruct temperature on the base of $\delta^{15}N$-fitting. It is visible that the deviation between the three different scenarios can be up to more than ±10% in the early-Holocene and decreases over time (fig. 4b). The deviations between the three scenarios have a minor influence on the modelled $\Delta$age during the Holocene (fig. 5c,d). Starting with a maximum difference of about 30 yr in the early-Holocene, the $\Delta$age difference decreases until 5 kyr b2k following the decrease in the difference between the accumulation-rate data. From the mid-Holocene until today the modelled $\Delta$age difference becomes less than 5 yr. The same is true for the spread in LID (fig. 5e). Here also the effect of the adjustment is visible, forcing the same firn-state for all accumulation-rate scenarios at the beginning of the reconstruction window (supplement sect. S2). Whereas the reconstructed temperatures when using the 50 km and 100 km accumulation-rate scenarios lead mainly to the same temperature trend, the reconstruction using the 200 km scenario shows a slightly larger cooling over the Holocene starting from about 7.5 kyr b2k (fig. 5f). This is exactly the point in time where the decrease in accumulation-rate for the 200 km scenario compared to the averaged scenario starts to accelerate (fig. 5b). As we do not see an equal (but opposite) behaviour for the 50 km scenario, we may have found a non-linear response between temperature and accumulation-rate change. From about 6 kyr to today, this faster cooling starting at 7.5 kyr levels out and leads to a constant offset of about 0.3 K between the 200 km solution and the two others, which is nearly 15% of the whole cooling trend of about 2 K referred to the warmest part of the reconstructed temperatures at around 7.8 kyr b2k. Besides this, the shapes and amplitudes of the faster signals of the long-term fractions (T > 500 yr) are highly comparable. The relatively large deviation of $\Delta$age between the scenarios in the early-Holocene leads to a slightly asynchronous behaviour of the short-term temperature variations (T < 500 yr, fig. 5g,h). But the shapes and amplitudes of the signals are independent from the deviations between the accumulation-rate scenarios. The decreasing deviation between the accumulation-rate scenarios and thereof between the modelled $\Delta$age over the Holocene leaves no differences between the short-term fractions of the reconstructed temperatures in the late-Holocene section (fig. 5h). To sum up, the deviation between the three different accumulation-rate scenarios do not have a major impact on the reconstructed temperature anomalies. The differences between the accumulation-rates lead to slightly different modelled $\Delta$age in the early-Holocene and to a 0.3 K larger cooling for the higher accumulation-rate scenario compared to the two other ones.

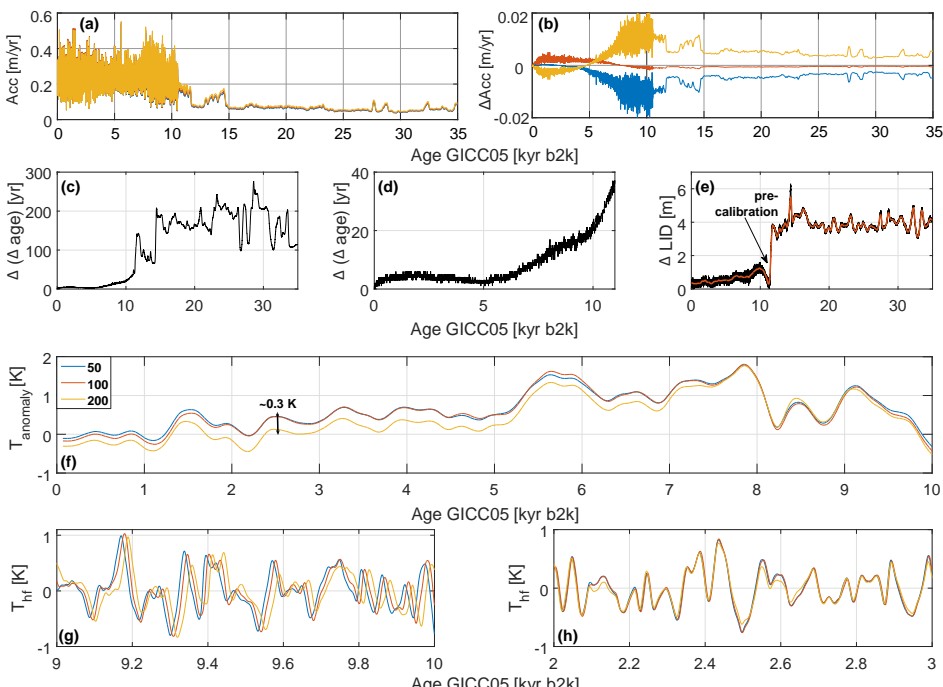

**Figure 5:** Influence of different accumulation-rate scenarios ((a): blue: 50 km, red: 100 km, yellow: 200 km) on the reconstruction. B: Deviation of each single scenario to the average of all three scenarios. (c): Maximum spread in modelled Δage using the three different accumulation-rate scenarios for the whole input (Holocene and spin-up). (d): Zoom-in for (a) for the Holocene part. (e): Maximum spread in the modelled Lock-In-Depth (LID), the pre-calibration leads to a convergence of the LID and therefore to the same gravitational background for the isotope signals. (f): Long-term temperature trend $T_{anomaly}$ (low-pass: > 500 yr) of the reconstructed temperatures modelled using the three accumulation-rate scenarios. (g): Short-term temperature signals $T_{hf}$ (high-pass: < 500yr) showing asynchrony in the early-Holocene, as a result of the spread in Δage. (h): Short-term temperature signals $T_{hf}$ (high-pass: < 500yr) showing synchrony from the mid- to late-Holocene, as a result of the decreasing spread in Δage and the decreasing difference between the accumulation-rates.

## 4 Results and discussion: temperature reconstruction

In this section we show and discuss the temperature estimates emerging when fitting the different gas-isotope targets independently ($\delta^{15}N$, $\delta^{40}Ar$, $\delta^{15}N_{excess}$) using the Schwander et al. (1997) and Goujon et al. (2003) firn-model (sects. 4.1. and 4.2.). All temperature estimates are shown as anomalies (relative to 11.3 kyr b2k). The modelled firn parameters (Δage, LID) for each target are shown in fig. S3. Additionally, a so-called hybrid solution is created using $\delta^{15}N$ together with $\delta^{15}N_{excess}$, as follows: The hybrid solution is created from the mean temperature solution of $\delta^{15}N$-fitting, low-pass filtered with a cut-off-period of 500 yr giving a long-term temperature trend. This long-term temperature trend is superimposed by adding high-frequency information calculated from $\delta^{15}N_{excess}$. The high-frequency temperatures are calculated by translating the mismatches of modelled – using the long-term temperature trend (from $\delta^{15}N$) – and measured $\delta^{15}N_{excess}$ data ($\Delta\delta^{15}N_{excess,hf}$) into temperature by using the temperature sensitivities $\Omega_N$ of $\delta^{15}N$ and $\Omega_{Ar}$ of $\delta^{40}Ar$ as follows:

$$\Delta T_{hybrid,hf}(t) = \frac{\Delta\delta^{15}N_{excess,hf}(t)}{\left(\Omega_N(t) - \Omega_{Ar}(t)/4\right)} \tag{3}$$

The hybrid solution is used to imitate the temperature reconstruction by Kobashi et al. (2017) but with a different strategy. Kobashi et al. (2017) fitted the temperature gradient over the diffusive firn-column $\Delta T_{firn}$ (eq. 4)





calculated from $\delta^{15}N_{excess}$. Due to the high relative uncertainty of $\delta^{15}N_{excess}$ together with the yearly calculation of surface temperature from the modelled bottom temperature values using the $\Delta T_{firn}$-integration-method (Kobashi et al., 2008a, 2010), strong drifts in the reconstructed temperature can occur (sect. 4.1.2). To overcome this issue, Kobashi et al. (2017) forces the $\delta^{15}N_{excess}$ temperature to also fit the general trend of $\delta^{15}N$ by allowing constant

shifts in $\Delta T_{firn}$ in nine certain time windows of 1500 yr length. This correction reduces the goodness of the $\delta^{15}N_{excess}$-fit in some parts of the time-series and adds additional uncertainty to the reconstructed temperature. Additionally, the correction can change millennial scale trends, because changing the mean $\Delta T_{firn}$ in a 1500°yr window directly changes the temperature trend in this section. Also, allowing sharp shifts between the windows can create short term temperature signals (jumps with 50-200 yr durations), which could be misinterpreted as real

temperature changes. As our method provides the possibility to know the long-term $T(\delta^{15}N)$ trend, it is interesting to compare the hybrid temperature solution to the Kobashi et al. solution, and to investigate the differences which should mainly occure due to the "window correction" method.

### 4.1 Temperature solution comparison

#### 4.1.1 Model comparison

The comparison of the fitting results using the Schwander et al. (1997) or the Goujon et al. (2003) firn-model (fig. 4a-d) on an individual fitting target reveals that except for $T(\delta^{15}N_{excess})$, the solutions gained by using the two models are highly comparable. They show high correlation (r > 0.9) in all considered periodic-time bands (supplement section S6, tabs. S3-S6). The absolute temperatures show an offset of about 2 K between the two models. This offset can be explained by the implemented convective zone in the Goujon et al. (2003) model. As

the convective zone lowers the height of the diffusive firn column, a colder temperature (compared to the Schwander et al. (1997) model) is needed, decelerating the densification and leading to the LID needed to fit the gravitational fraction of the gas-isotope data. The temperature differences between the estimates using both models are not fully constant. Besides the offset we find mean differences (2σ) of 0.62 K for $T(\delta^{15}N)$ and 0.73 K for $T(\delta^{40}Ar)$, which are partly driven by remaining mismatches of the isotope fits. In the early-Holocene (9.5 kyr to

11.5 kyr) the $T(\delta^{15}N)$ and $T(\delta^{40}Ar)$ reconstructions using the Goujon et al. (2003) firn-model show a faster warming, leading to slightly higher temperature anomalies compared to the Schwander et al. (1997) model estimates. The rest of the time-series shows the reverse behaviour. Here the Schwander et al. (1997) model estimates are slightly warmer than Goujon et al. (2003) model estimates. The fact that the trend in the time-series of the differences between the Schwander et al. (1997) and Goujon et al. (2003) model temperature solution

follows the trend in the reconstructed temperature anomalies is pointing to a temperature sensitive fraction of these differences. An explanation for this could be that the densification itself is slightly temperature depended. The 2 K colder absolute temperature in the Goujon et al. (2003) model inputs leads to a different densification pathway compared to the Schwander et al. (1997) model runs. Nevertheless, both models provide highly comparable temperature estimates if temperature anomalies are considered.

#### 4.1.2 Comparison of $\delta^{15}N$ and $\delta^{40}Ar$ reconstruction

The comparison between the $\delta^{15}N$ and $\delta^{40}Ar$ based temperatures (fig. 4a,b and fig. S6) shows that the general trends between both reconstructions are very similar. Also, the shapes of many of the shorter-term temperature features are in a good agreement, but $T(\delta^{40}Ar)$ points to higher amplitudes of these anomalies. We find high correlations (supplement sect. S6, tabs. S3-S6) for the low pass filtered data (cut-off: 50 yr, r = 0.96) and the multi-





millennial band (r = 0.94), which was expected due to the high agreement in the long-term trends between both temperatures. The multi-centennial band shows a lower but still high correlation (r = 0.87). In the multi-decadal band, the correlation between T($\delta^{15}$N) and T($\delta^{40}$Ar) is weak (r = 0.69, lag = -4 yr or r = 0.67 for lag = 0 yr), and equals the correlation between the measured isotopes ($\delta^{15}$N and $\delta^{40}$Ar) in the same band with r = 0.68. This result

is not surprising because: (i) The high-frequency fraction of the reconstructed temperatures is directly calculated from the high-frequency fraction of the respective isotope targets (Döring and Leuenberger, 2018) and (ii) the accumulation-rate input has only a minor effect on the reconstructed temperatures in multi-decadal band. The mean offset between T($\delta^{15}$N) and T($\delta^{40}$Ar) over the whole time-series is 0.28 K and the standard deviation (2$\sigma$) of the differences is 1.00 K for the maximum resolution case (mean resolution of the isotope data is 17.8 yr). In the

early to mid (6.4 kyr-11.5 kyr) and late-Holocene (0.07 kyr-1.3 kyr b2k), the $\delta^{15}$N reconstruction leads to a slightly higher absolute temperature compared to T($\delta^{40}$Ar) with mean-offsets of about 0.46 K for the early- to mid-Holocene and 0.61 K for the late-Holocene, whereas the rest of the temperature time-series is showing a similar trend (1.3 kyr-6.4 kyr b2k, mean offset: 0.06 K). The lower temperatures in T($\delta^{40}$Ar) in the early- to mid- and late-Holocene can be explained by an enrichment in $\delta^{40}$Ar due to gas-loss still remaining after the applied corrections.

The LID estimates of both reconstructions are in good agreement (fig. S6b). The differences between them vary in a narrow band of -2 m to 1 m driven by the temperature differences between the reconstructions. The comparison of modelled firn temperature gradients $\Delta T_{firn}$ (fig. S6c) of the $\delta^{15}$N- and $\delta^{40}$Ar-fits shows a high agreement in the general trends and in the shapes of the shorter-term features. The differences between them are less than ±1 K in most of the case. Additionally, $\Delta T_{firn,meas}$ calculated directly from the measured isotope data (dotted line, meas)

together with its maximum (1$\sigma_{max}$) and minimum (1$\sigma_{min}$) uncertainty, is compared to the modelled estimates. $\Delta T_{firn,meas}$ was calculated analogous to Kobashi et al. (2010, 2011, 2017) from $\delta^{15}$N$_{excess}$ according to:

$$\Delta T_{firn,meas} = \frac{\delta^{15}N_{excess}}{\left(\Omega_N - \frac{\Omega_{Ar}}{4}\right)} = \frac{\delta^{15}N - \frac{\delta^{40}Ar}{4}}{\left(\Omega_N - \frac{\Omega_{Ar}}{4}\right)} \qquad (4)$$

The uncertainties (1$\sigma_{max}$, 1$\sigma_{min}$) of $\Delta T_{firn,meas}$ were calculated using Gaussian-error-propagation on eq. 4 together with the uncertainties of $\delta^{15}$N and $\delta^{40}$Ar as stated in tab. S1. If we compare the modelled $\Delta T_{firn}$ to $\Delta T_{firn,meas}$ we

find a good agreement of the general trend in the late- to mid-Holocene (1.3 kyr-6.4 kyr b2k), which is exactly the part in time where also the trends in T($\delta^{15}$N) and T($\delta^{40}$Ar) are showing the smallest offset. In the early- to mid-Holocene (6.4 kyr-11.5 kyr), $\Delta T_{firn}$ modelled from $\delta^{15}$N and $\delta^{40}$Ar significantly exceeds $\Delta T_{firn,meas}$, which can be a sign for systematic too high $\delta^{40}$Ar in this section, reducing $\delta^{15}$N$_{excess}$ and $\Delta T_{firn,meas}$. The same is true for the late-Holocene (0.07 kyr-1.3 kyr b2k). Comparing the amplitudes of the faster signals of the measured and modelled

$\Delta T_{firn}$ shows that the modelled signals underestimates the amplitudes of $\Delta T_{firn,meas}$, which leads to the assumption that $\Delta T_{firn,meas}$ and therefore $\delta^{15}$N$_{excess}$ is potentially more driven by noise in the isotope data than temperature.

### 4.1.2 Comparison of $\delta^{15}$N and $\delta^{15}$N$_{excess}$ reconstruction

Precise fitting of $\delta^{15}$N$_{excess}$ and therefore $\Delta T_{firn}$ (fig. 4d and fig. S7) results in a distinct temperature regime compared to T($\delta^{15}$N) and T($\delta^{40}$Ar). In the early-Holocene (9 kyr-11.5 kyr) the fitting of $\delta^{15}$N$_{excess}$ leads to a flat

temperature with nearly no trend. This is not only in disagreement with T($\delta^{15}$N) or T($\delta^{40}$Ar) but also with the reconstructions from Kobashi et al. (2017) and Buizert et al. (2018) for the GISP2 site (fig. 6). The flat temperatures in this section when precisely fitting $\delta^{15}$N$_{excess}$ is the result of too low $\delta^{15}$N$_{excess}$ and therefore $\Delta T_{firn}$, which is driven by too high $\delta^{40}$Ar in that section. In the late-Holocene, T($\delta^{15}$N$_{excess}$) shows a large cooling of about -3.6 K/kyr (0.16 kyr-1.25 kyr), which is highly unrealistic, pointing also to too low $\delta^{15}$N$_{excess}$ and $\Delta T_{firn}$. Also the



$\delta^{15}N_{excess}$-fit of Kobashi et al. (2017) shows large disagreement between their modelled $\delta^{15}N_{excess}$ and the measured ones during this sections (see supplementary information fig. S3 in Kobashi et al. (2017)), which means that the quality of the $\delta^{15}N_{excess}$-fit has to be reduced significantly to extract a potentially meaningful temperature estimate. As we have used the corrected $\delta^{40}Ar$ to calculate $\delta^{15}N_{excess}$ and $\Delta T_{firn}$, we have to argue that the presently available

correction (Kobashi et al., 2015b, 2017) is non-sufficient, especially for the late- and early-Holocene data. This result is somehow surprising as it was argued in Buizert et al. (2018) that the influence of possible gas-loss on $\delta^{40}Ar$ is most severe within the bubble-clathrate transition zone (about 800 m to 1500 m depth of the GISP2 core, equals 3.8-9.3 kyr BP ice age, see supplement p. X-2 in Buizert et al. (2018)). In the following we compare two of the major mid-Holocene cooling trends in $T(\delta^{15}N_{excess})$ with the simultaneous trends in $T(\delta^{15}N)$. In the time

range 2 kyr to 4.8 kyr b2k, where the trends in $T(\delta^{15}N)$ and $T(\delta^{40}Ar)$ are highly comparable, $T(\delta^{15}N_{excess})$ shows a cooling rate of -0.52 K/kyr. The cooling trend of $T(\delta^{15}N)$ in the same time range is -0.18 K/kyr and therefore about 3 times smaller. In the second time range from 6.3 kyr to 8.1 kyr, where $T(\delta^{15}N)$ and $T(\delta^{40}Ar)$ show a significant offset, the cooling in $T(\delta^{15}N_{excess})$ exceeds the cooling in $T(\delta^{15}N)$ by a factor of about 2.5 (-1.58 K/kyr for $T(\delta^{15}N_{excess})$, -0.62 K/kyr for $T(\delta^{15}N)$). Also, these results could be explained by too low $\delta^{15}N_{excess}$ and $\Delta T_{firn}$. Next,

we compare the correlations of $T(\delta^{15}N_{excess})$ with $T(\delta^{15}N)$ and $T(\delta^{40}Ar)$ in the same periodic-time bands as it was done in the previous section (supplement sect. S6, tabs. S3-S6). In all periodic-time bands the correlation of $T(\delta^{15}N_{excess})$ with $T(\delta^{15}N)$ as well as $T(\delta^{15}N_{excess})$ with $T(\delta^{40}Ar)$ is weak. The highest correlation between $T(\delta^{15}N_{excess})$ and $T(\delta^{15}N)$ was found for the multi-millennial time-band (tab. S4) with r = 0.61 for the best fit $T(\delta^{15}N_{excess})$ solution and r = 0.68 for the averaged $T(\delta^{15}N_{excess})$ solution. The correlation with $T(\delta^{40}Ar)$ is even

weaker (r = 0.48, best fit; r = 0.54, mean solution). In the multi-decadal periodic-time band (tab. S6) we find a weak negative correlation between $T(\delta^{15}N_{excess})$ and $T(\delta^{40}Ar)$ with

r = -0.41 and r = -0.36 for the best fit and the mean $T(\delta^{15}N_{excess})$ solution, respectively. The result for the multi-decadal time band can be explained, since the multi-decadal oscillations in $T(\delta^{15}N_{excess})$ are mainly driven by $\delta^{40}Ar$ with less influence from $\delta^{15}N$ due to higher variability of $\delta^{40}Ar/4$ compared to $\delta^{15}N$.

**4.2 Final uncertainty of reconstructed temperature**

Using all information from the previous sections we can calculate an overall limit of the mean uncertainty for reconstructed temperatures using the following equation:

$$(2\sigma_T)^2 = \left(\frac{2\sigma_{miss}}{\Omega_X}\right)^2 + \left(2\sigma_{rep}\right)^2 + \left(\frac{2\sigma_{meas}}{\Omega_X}\right)^2 + (2\sigma_{model})^2 \tag{5}$$

$\sigma_T$ is the uncertainty of the reconstructed temperature, $\sigma_{miss}$ is the remaining mean mismatch after fitting the isotope

values (sect. 3.1.1), $\sigma_{rep}$ is the fraction of uncertainty due to the reproducibility of the fitting method (sect. 3.1.1), $\sigma_{meas}$ is the analytical uncertainty of the measured data (tab. S1) and $\sigma_{model}$ is the standard deviation of the differences of the temperature anomalies between the used models (sect. 4.1.1). $\Omega_x$ is the thermal diffusion sensitivity of the respective isotope species. For $T(\delta^{15}N)$ the calculated uncertainty is $2\sigma_T = 0.80...0.88$ K, the range is due to the minimal or maximal analytical uncertainty of the measured data. The uncertainty of the $\delta^{40}Ar$

reconstruction is $2\sigma_T = 0.87...1.81$ K. This final uncertainty is attributed to each single temperature point in time (the mean data resolution was 17.8 yr), so a smoothing or running mean calculation will decrease the uncertainty due to the averaging over a certain amount of points. For example, a smoothing with 100 yr cut-off will reduce the uncertainty with a factor of $1/n^{1/2} = 1/(100 \text{ yr}/17.8 \text{ yr})^{1/2} = 0.42$ as it equals an averaging over 5.6 points. This is important to keep in mind when discussing filtered versions of these reconstructed temperatures. For $T(\delta^{15}N_{excess})$

we are not able to provide a final uncertainty, as we do not understand the reason for the different behaviour when





fitting the data with the two models (stable Goujon- vs. unstable Schwander-solutions). However, as fitting of $\delta^{15}N_{excess}$ leads to highly distinct temperature estimates compared to $T(\delta^{15}N)$ or $T(\delta^{40}Ar)$ and also when compared to other reconstructions (comparison of fig. 4d with fig. 6a), we do not recommend to use $T(\delta^{15}N_{excess})$ for any climatic interpretation yet. As $\delta^{15}N$ is easier to measure due to the higher abundance of nitrogen in air and less

susceptible to gas-loss induced fractionation (Huber et al., 2006a), we argue that $T(\delta^{15}N)$ provides the most robust temperature estimate compared to $T(\delta^{40}Ar)$ and especially $T(\delta^{15}N_{excess})$.

### 4.3 Comparison of $T(\delta^{15}N)$ with the reconstructions of Kobashi et al. 2017 and Buizert et al. 2018

Figure 6 shows the comparison of $T(\delta^{15}N)$ (blue lines) with the reconstructions of Buizert et al. (2018) (red lines) and Kobashi et al. (2017) (black lines) and the comparison between the hybrid-temperature (green lines, fig. 6b)

with the Kobashi et al. (2017) solution. As stated before, the temperature reconstructions of Kobashi et al. (2017) were conducted using $\delta^{15}N_{excess}$ to obtain temperature, which is in our opinion problematic due to the high relative uncertainty of $\delta^{15}N_{excess}$ and the systematic offsets to too high $\delta^{40}Ar$. Buizert et al. (2018) use the calibration of water-stable-isotope $\delta^{18}O_{ice}$ to fit the long-term trend of $\delta^{15}N$ for the early- to mid-Holocene (until 4 kyr b2k). From 4 kyr to today they use the temperature from Kobashi et al. (2017) superimposed with a larger cooling trend

(fig. S8). First, we compare the variance of the temperatures (tab. 2) for two time-windows (0.5-4.0 kyr b2k and 4.0-11.2 kyr b2k) and two periodic-time bands (bands: 100 yr to 500 yr and 500 yr to 4 kyr). As the temporal resolution of the Buizert et al. (2018) estimate is 20 yr, we resampled our data and the Kobashi et al. (2017) data to the same grid before band-pass filtering. Also, we cut out the 8.2k-event, as it dominates the variance of the temperature data. In the early- to mid-Holocene (4.0-11.2 kyr b2k, w/o 8,2k-event), the standard deviation (2σ) of

$T(\delta^{15}N)$ in the 100-500 yr periodic-time band is 0.47 K which is nearly equal to the Buizert et al. (2018) reconstruction with a value of 0.49 K. The Kobashi et al. (2017) reconstruction has a more than double as large variance in that section with 2σ = 1.17 K. For the mid- to late-Holocene (0.5-4.0 kyr b2k), the standard deviation of $T(\delta^{15}N)$ is about 20% smaller (2σ = 0.37 K) than for the early- to mid-Holocene (4.0-11.2 kyr). The variance of the Kobashi et al. (2017) reconstruction is slightly smaller (17%, 2σ = 0.97 K) during this time. The Buizert et al.

(2018) reconstruction nearly equals the Kobashi et al. (2017) estimate here, with 2σ = 1.00 K. This is not unexpected as Buizert et al. (2018) use the Kobashi et al. (2017) data with slight modifications. But it is not quite reasonable that a doubling of the variance between the two parts of the Holocene is realistic. For longer periodicities (band: 0.5-4.0 kyr) we also see a reduction of the variance for the late-Holocene compared to the early-Holocene in all three reconstructions. During the early- to mid-Holocene the variance of $T(\delta^{15}N)$ and the

Buizert et al. (2018) reconstruction are in good agreement ($2\sigma_{T(\delta15N)} = 0.58$ K, $2\sigma_{Buizert} = 0.65$ K), whereas the Kobashi et al. (2017) estimate is pointing to higher variability ($2\sigma_{Kobashi} = 1.26$ K), an equal behaviour as it was found for the faster periodicities. For the mid- to late-Holocene, $T(\delta^{15}N)$ shows the smallest variability (2σ = 0.35 K) compared to the Buizert et al. (2018) (2σ = 0.58 K) and Kobashi et al. (2017) (2σ = 0.81 K) reconstructions. We conducted the same analysis for the accumulation-rate data and for the gas-isotope data

(tab. 3). Especially for the 0.5-4.0 kyr periodic-time band, we find an equal reduction of the data variance between the early- to mid- and the mid- to late-Holocene. For $\delta^{15}N$ the standard deviation (2σ) in the early- to mid-Holocene is about 52% higher as in the mid- to late-Holocene, 34% for $\delta^{40}Ar$ and 41% for $\delta^{15}N_{excess}$, respectively. The accumulation-rate data show this deviation of the variance between the two time-sections in this periodic-time band with a reduction of 46%. For the faster periodicities (band-pass 100-500 yr), the gas-isotope data also shows that behaviour, but with less disagreement between the two time-sections. In contrast, the accumulation-rate data






show a slightly higher variance in the mid- to late-Holocene if compared to the early- to mid-Holocene. Based on those findings we conclude that the difference in the variability of our temperature estimates between the early- to mid- and the mid- to late-Holocene is a direct result of that behaviour of the gas-isotope target data and the accumulation-rate input. Interestingly, the difference between the Buizert et al. (2018) and Kobashi et al. (2017)

estimates (fig. S8) for the time section 0.5-4.0 kyr b2k, points to the necessity of modifications on the Kobashi et al. (2017) estimate when used in the study of Buizert et al. (2018). A reason for that could be the use of a different firn-model or possible memory effects occurring due to the differences in the temperature estimates between the Buizert et al. (2018) and Kobashi et al. (2017) estimates in the early- to mid-Holocene. Figure 6b shows the comparison between the Kobashi et al. (2017) estimate and our hybrid temperature. It is obvious that both estimates

agree well, especially for the faster features. In four time-sections we find larger offsets. These sections are: 9-10.5 kyr, shortly after the 8.2k-event (6.6-8.1 kyr), 5.3-6.1 kyr and 0.07-1.8 kyr. All sections start with a fast warming or cooling trend in the Kobashi et al. (2017) estimate with duration of about 100-200 yr. It is highly probable that these shifts are introduced by the "window correction method" used by Kobashi et al. (2017). In our view, this correction method is highly critical as the choice of the window-length, the window positions and the

found offsets are arbitrary, but on the other side crucial for the reconstruction. Additionally, we correlated the Buizert et al. (2018) and Kobashi et al. (2017) temperature estimates with each other and with our data after low-pass filtering (cop = 50 yr) and in all three investigated periodic-time bands (multi-decadal, multi-centennial, multi-millennial, tab. S7). For the Buizert et al. (2018) reconstruction the correlations were calculated only for the early- to mid-Holocene values (4.0-11.5 kyr b2k), whereas the correlations between our and the Kobashi et al.

(2017) estimates were calculated for 0.5-11.5 kyr b2k. In the low-pass filtered case (general trend) the Buizert et al. (2018) reconstruction shows the highest correlations with $T(\delta^{15}N)$ and $T(\delta^{40}Ar)$ with $r > 0.9$ and a correlation of $r = 0.82$ with the Kobashi et al. (2017) estimate, due to the high agreement in the general trend between the three studies. The comparison between the Kobashi et al. (2017) temperature and our estimates has the highest correlation with the hybrid temperatures ($r = 0.87$, $r = 0.83$), and $T(\delta^{15}N)$ ($r = 0.81$) in that case. Besides the general

trend, the correlations become reduced for faster oscillations. In all periodic-time bands the Buizert et al. (2018) reconstruction has the highest correlation with $T(\delta^{15}N)$ with $r_m = 0.67$, $r_c = 0.61$ and $r_d = 0.30$ for multi-millennial, multi-centennial and multi-decadal signals, respectively. The correlation coefficients between the Buizert et al. (2018) and the Kobashi et al. (2017) reconstruction are $r_m = 0.52$ ($p = 0.03$), $r_c = 0.48$ and $r_d = 0.05$ ($p = 0.25$). That finding implies that the agreement between the Buizert et al. (2018) $\delta^{18}O_{ice}$-based and our $\delta^{15}N$-based

reconstruction is generally higher as for the Buizert et al. (2018) reconstruction compared to the $\delta^{15}N_{excess}$-based one of Kobashi et al. (2017). On the other hand, the correlations are relatively weak, especially for the multi-decadal band.





| time section [kyr b2k] | Band-pass: 100 yr-500 yr | | | Band-pass: 0.5 kyr-4.0 kyr | | |
|---|---|---|---|---|---|---|
| | $2\sigma$ [K] $T(\delta^{15}N)$ | $2\sigma$ [K] Buizert 2018 | $2\sigma$ [K] Kobashi 2017 | $2\sigma$ [K] $T(\delta^{15}N)$ | $2\sigma$ [K] Buizert 2018 | $2\sigma$ [K] Kobashi 2017 |
| 0.5-11.2 w/o 8.2k-event | 0.44 | 0.72 | 1.10 | 0.51 | 0.63 | 1.12 |
| 0.5-4.0 | 0.37 | 1.00 | 0.97 | 0.35 | 0.58 | 0.81 |
| 4.0-11.2 w/o 8.2k-event | 0.47 | 0.49 | 1.17 | 0.58 | 0.65 | 1.26 |

Table 2: Standard deviations ($2\sigma$) of temperature estimates of $T(\delta^{15}N)$ (this study), Buizert et al. (2018) and Kobashi et al. (2017) when band-pass filtered for two periodic-time bands and calculated without the 8.2k-event.

| time section [kyr b2k] | Band-pass: 100 yr-500 yr | | | | Band-pass: 0.5 kyr-4.0 kyr | | | |
|---|---|---|---|---|---|---|---|---|
| | $2\sigma$ [permeg] $\delta^{15}N$ | $2\sigma$ [permeg] $\delta^{40}Ar$ | $2\sigma$ [permeg] $\delta^{15}N_{excess}$ | $2\sigma$ [mm/yr] acc | $2\sigma$ [permeg] $\delta^{15}N$ | $2\sigma$ [permeg] $\delta^{40}Ar$ | $2\sigma$ [permeg] $\delta^{15}N_{excess}$ | $2\sigma$ [mm/yr] acc |
| 0.5-11.2 w/o 8.2k-event | 7.2 | 6.9 | 5.2 | 11.7 | 6.6 | 5.5 | 3.0 | 8.2 |
| 0.5-4.0 | 6.6 | 5.9 | 3.9 | 13.0 | 3.7 | 4.0 | 2.0 | 5.1 |
| 4.0-11.2 w/o 8.2k-event | 7.5 | 7.3 | 5.7 | 11.0 | 7.7 | 6.1 | 3.4 | 9.4 |

Table 3: Standard deviations ($2\sigma$) of gas-isotope measured data and accumulation-rates when band-pass filtered for two periodic-time bands and calculated without the 8.2k-event.




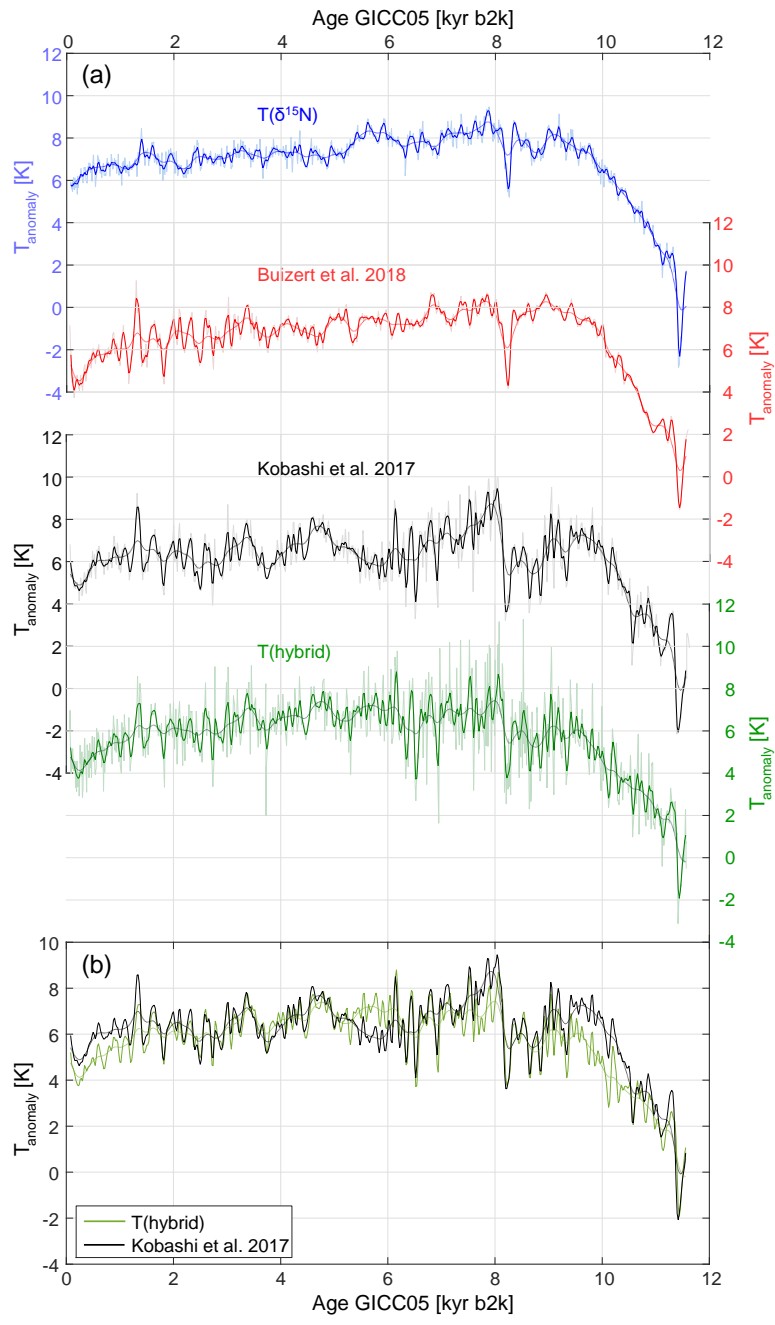

**Figure 6: (a) Comparison of T(δ¹⁵N) (blue lines) and T(hybrid) (green lines) modelled using the Schwander et al. model with the temperature reconstructions (anomalies rel. to 11.500 kyr b2k) for the GISP2 site from Buizert et al. 2018 (red lines) and Kobashi et al. 2017 (black lines). Thin lines show unfiltered data, thick lines show low-pass filtered data using cop = 100 yr, dotted lines show low-pass filtered data using cop = 500 yr. (b) Comparison between the reconstruction from Kobashi et al. 2017 (black lines) and T(hybrid) (lines) in the same plot.**

## 5 Conclusion

In this study we applied the Döring and Leuenberger (2018) gas-isotope fitting-algorithm to Holocene $\delta^{15}N$, $\delta^{40}Ar$ and $\delta^{15}N_{excess}$ data measured on the GISP2 ice core (Kobashi et al., 2008b) using two state of the art firn-densification and heat-diffusion models (Goujon et al., 2003; Schwander et al., 1997). The results of this study are

summarized as follows:

*Signal-to-noise study:*

As starting point, a signal-to-noise (SNR) study was conducted, investigating the suitability of the three gas-isotope tracers ($\delta^{15}N$, $\delta^{40}Ar$ and $\delta^{15}N_{excess}$) for temperature reconstructions in context of the in Kobashi et al. (2008b) stated measurement uncertainty. It was shown that $\delta^{15}N$ is most favoured to reconstruct Holocene temperature due to its

higher SNR compared to $\delta^{40}Ar$ and $\delta^{15}N_{excess}$, especially for multi-decadal to multi-centennial signals.

*Gas-isotope fitting results and uncertainties:*

To evaluate the performance of the Döring and Leuenberger (2018) gas-isotope fitting-algorithm on measured Holocene data and to constrain the uncertainty of the reconstructed temperatures, the reproducibility of the fitting algorithm was tested using the Schwander et al. (1997) firn-model. The results are showing excellent performance

for $\delta^{15}N$ and $\delta^{40}Ar$-fits. The reproducibility between ten runs of the fitting algorithm was quantified to be (mean ± 2σ) 1.68 ± 1.14 permeg for $\delta^{15}N$, 2.58 ± 1.77 permeg for $\delta^{40}Ar/4$, and 1.28 ± 2.14 permeg for $\delta^{15}N_{excess}$ for each single point in time. The translation in temperature also shows good reproducibility with (0.17 ± 0.12 K) for $T(\delta^{15}N)$ and (0.26 ± 0.18 K) for $T(\delta^{40}Ar)$. In contrast, the spread for $\delta^{15}N_{excess}$ temperature was about ten times higher (2.04 ± 1.90 K) due to the loss of information about the firn-column-height when calculating $\delta^{15}N_{excess}$,

which leads to a variety of different possible temperature solutions fitting $\delta^{15}N_{excess}$ with the same precision. In addition, the contribution of remaining misfits to the uncertainty budget of the reconstructed temperatures was quantified. We reach final misfits (2σ) of 3.7 permeg for $\delta^{15}N$, 2.8 permeg for $\delta^{40}Ar/4$, and 2.1 permeg for $\delta^{15}N_{excess}$ using the Schwander et al. (1997) model, and 6.0 permeg, 7.0 permeg and 2.0 permeg for $\delta^{15}N$, $\delta^{40}Ar/4$ and $\delta^{15}N_{excess}$ respectively using the Goujon et al. (2003) model. Additionally, the influence of three different

accumulation-rate estimates on the temperature reconstructions was investigated. The accumulation-rate uncertainty leads to asynchronous multi-decadal temperature signals due to deviations in the modelled Δage regimes, with maximum differences of 30 yr in the early-Holocene but keeping the amplitudes of the signals unchanged. From the mid-Holocene until today, the modelled Δage difference becomes less than 5 yr. For longer term temperature trends, the higher accumulation-rate scenario leads to a 0.3 K larger cooling compared to the

lower and intermediate accumulation-rate scenarios.

*Comparison of model results:*

Next, we compared the temperature estimates calculated by fitting $\delta^{15}N$, $\delta^{40}Ar$ and $\delta^{15}N_{excess}$ with each other. We found significant and high correlation between $T(\delta^{15}N)$ and $T(\delta^{40}Ar)$ for multi-centennial and multi-millennial signals, but $T(\delta^{40}Ar)$ points to higher amplitudes for some of these temperature anomalies. For multi-decadal

signals the correlation between $T(\delta^{15}N)$ and $T(\delta^{40}Ar)$ is weak but still significant and equals the correlation between the isotope data ($\delta^{15}N$ and $\delta^{40}Ar$). The comparison of the temperature gradient over the diffusive firn column $\Delta T_{firn}$ modelled from $T(\delta^{15}N)$ and $T(\delta^{40}Ar)$ with the $\Delta T_{firn,meas}$ – calculated from the measured $\delta^{15}N_{excess}$ – documents a good agreement of the general trend in the late- to mid-Holocene (1.3 kyr-6.4 kyr b2k). However, in the early to mid-Holocene (6.4 kyr-11.5 kyr) and the late-Holocene (0.07 kyr-1.3 kyr b2k), $\Delta T_{firn}$ modelled from $\delta^{15}N$ and

$\delta^{40}Ar$ significantly exceeds $\Delta T_{firn,meas}$, which might be an indication for systematic too high $\delta^{40}Ar$ in this section, potentially caused by gas-loss induced fractionation of $\delta^{40}Ar$. The temperature calculated by fitting $\delta^{15}N_{excess}$ differs



significantly from the coherent $T(\delta^{15}N)$ and $T(\delta^{40}Ar)$, especially for the early- and late-Holocene. The analysis of these differences also suggests that the $\delta^{40}Ar$ values are too high in these sections. The correlations between $T(\delta^{15}N_{excess})$ with $T(\delta^{15}N)$ or $T(\delta^{40}Ar)$ are weak for all three analysed periodic-time bands (multi-decadal, multi-centennial, multi-millennial). For multi-decadal signals we find a weak negative correlation between $T(\delta^{15}N_{excess})$

and $T(\delta^{40}Ar)$ and no correlation between $T(\delta^{15}N_{excess})$ and $T(\delta^{15}N)$, which implies that the multi-decadal oscillations in $T(\delta^{15}N_{excess})$ are mainly driven by $\delta^{40}Ar$ with less influence from $\delta^{15}N$ due to a higher noise contribution on $\delta^{40}Ar/4$ compared to $\delta^{15}N$. In addition, we calculated the slope between $\delta^{15}N$ and $\delta^{40}Ar$ using geometric-mean-regression for all three periodic-time bands. Especially for multi-decadal signals the slope significantly underestimates the theoretical value (calculated using the empirical derived thermal sensitivities of

$\delta^{15}N$ and $\delta^{40}Ar$) by 53%. These results are pointing to too high $\delta^{40}Ar$ that may be influenced by noise or fractionation which is not captured by the firn-models (e.g. gas loss). For the multi-centennial and the multi-millennial bands, the slope equals the theoretical expectation.

*Comparison of model results of two different firn-models:*

Next, we compared reconstructed temperatures obtained by using the Schwander et al. (1997) firn-model with the

solutions of the Goujon et al. (2003) model. For $T(\delta^{15}N)$ and $T(\delta^{40}Ar)$ the temperature estimates show high correlation (r > 0.9) in all considered periodic-time bands (multi-decadal, multi-centennial, multi-millenial) and in amplitudes and shapes of many of the shorter-term features. In the early-Holocene (9.5 kyr to 11.5 kyr) the $T(\delta^{15}N)$ and $T(\delta^{40}Ar)$ reconstructions using the Goujon et al. (2003) firn-model show a faster warming, leading to slightly warmer temperature anomalies compared to the Schwander et al. (1997) model estimates. The rest of the time-

series shows the reverse behaviour. Here the Schwander et al. (1997) model estimates are slightly warmer than the Goujon et al. (2003) model estimates. The result that the difference in the temperature anomalies between both models follows the general trend of the temperature anomalies themselves, is pointing to a temperature dependence which is attributed to the temperature dependence of the densification due to the difference in absolute temperatures of about 2 K. The variance of the differences ($2\sigma$) between the temperature anomalies obtained by

using both models were quantified to be 0.62 K for $T(\delta^{15}N)$ and 0.73 K for $T(\delta^{40}Ar)$.

*Uncertainty estimation:*

Using all results presented in this study we estimated the mean uncertainty for $T(\delta^{15}N)$ and $T(\delta^{40}Ar)$. The final uncertainty budget of the reconstructed temperature anomalies is dependent on four terms: (i) the mismatch between the measured and modelled isotope data; (ii) the reproducibility of the isotope fits; (iii) the measurement

uncertainty of the isotope data; and (iv) the difference between the temperature estimates using different firn-models (here between two firn-models). Adding up these terms leads to a final uncertainty of $2\sigma_T = 0.80...0.88$ K for $T(\delta^{15}N)$, and $2\sigma_T = 0.87...1.81$ K for $T(\delta^{40}Ar)$.

*Comparison to other published temperature reconstructions:*

Finally, we compared our temperature estimates to temperature reconstructions for the GISP2 site of Buizert et al.

(2018) and Kobashi et al. (2017). First, the variance of the temperature anomalies was analysed in two periodic-time bands (100-500 yr and 0.5-4 kyr). We found a high agreement of the variance between $T(\delta^{15}N)$ and the Buizert et al. (2018) estimate in both considered bands for the time 4.0-11.5 kyr b2k. In contrast, the variance of the Kobashi et al. (2017) temperature anomalies is nearly twice as high as our $T(\delta^{15}N)$ or the Buizert et al. (2018) estimate. Interestingly, all three reconstructions are pointing to a decrease of the variance in the mid- to late-

Holocene (0.5-4.0 kyr b2k) compared to the early- to mid-Holocene (4.0-11.5 kyr b2k) for multi-centennial to multi-millennial signals. For our reconstruction this result is attributed to an equal behaviour of the gas-isotope



and the accumulation-rate data. Finally, we compared the correlations between the three reconstructions for three periodic-time bands (multi-decadal, multi-centennial, multi-millennial). We find generally higher agreement in all bands between our $\delta^{15}$N-based reconstruction and the $\delta^{18}$O- and $\delta^{15}$N-based reconstruction of Buizert et al. (2018) as between our T($\delta^{15}$N) estimate and the Kobashi et al. (2017) reconstruction.

## 5   Competing interests

The authors declare that they have no competing financial interests.

### Acknowledgement

We would like to thank Gunnar Jansen for his IT support. This study was financed by the Swiss National Science Foundation through the icoCEP project (172550) as well as the project Climate and Environmental Physics (159563).

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
