# Peer review of "Comparison of Holocene temperature reconstructions based on GISP2 multiple-gas-isotope measurements"

_Climate of the Past, 2020_

## Short Comment (SC1) · 27 Oct 2020

Takuro Kobashi (kobashi.takuro@nies.go.jp)

Doring and Lueebergre conducted a study for calculating temperatures from d40Ar and d15N in trapped air in an ice core. They investigated various calculation methods and provided further understanding on calculating temperatures. The resubmitted manuscript has been improved. However, critical points are ignored for the sake of author's favor as pointed below. The authors need to consider seriously these points before publication as Greenland temperature is a highly important climate variable to understand the future consequences of increasing atmospheric CO2. Some of the

comments are the same as I wrote for the earlier version.

Main comments 1. d15N and d40Ar in ice cores are believed to be caused by gravitational fraction and thermal fractionation in firn. The magnitude of fractionation can be theoretically calculated according to laboratory calibration. Therefore, by measuring d15N and d40Ar in trapped air in ice cores, we can immediately know past temperature gradients in the top and bottom of firn (ïĄĎT) and “firn column depth (Dfirn)”. Using these data, Kobashi et al. (2010) invented a method to calculate surface temperature by integrating ïĄĎT using a firn-densification model. However, it was found that firn model outputs show firn depth cannot change as predicated by the isotopes (Dfirn). dNexcess (d15N-d40Ar/4) is highly robust temperature proxy as the calculation eliminate possible gas loss fractionation in the firn or from ice cores (Kobashi et al., 2008). Kobashi et al hypothesized that firn models are not capable to capture actual multidecadal temperature variability, and they produced Holocene temperatures avoiding utilization of firn model's densification process (Kobashi et al., 2017). On the other hand, Doring and Leuenberger hypothesize that firn model densification for a multidecadal scale is correct because the two firn models agree. Then, they calculate temperatures with the firn models. Therefore, two temperature calculations are based on two different hypotheses. As Doring and Leuenberger are not succeeding to reject the hypothesis of Kobashi et al. (2017), they provide only a hypothesis for Greenland temperature calculation, which is not fully tested with borehole temperature, oxygen isotopes of ice, or other climate proxies. 2. Kobashi et al. (2017) temperature is more advanced as it has been tested with various temperature proxies. Also, Holocene temperature variability (< 10,000 years) of Kobashi et al mostly arises from un-corrected dNexcess (orïĂăïĄĎT) (see Figure S2 in Kobashi et al. (2017)). Therefore, the different corrections such as time frame or argon correction has only minor effects on the calculated Greenland temperature. 3. Reconstructed temperature data using d15N, d40Ar, dNexcess, and hybrid should be submitted for others to securitize the results. 4. Authors did not demonstrate consistency with borehole temperature data. This is critical for the Holocene Greenland temperature from ice cores. Borehole temperature records

are arguably the most important physical evidence of past temperature. The Goujon model calculate borehole temperature for each realization. Authors must show their borehole temperature results in comparison to the GISP borehole temperature record. 5. Authors also must show a plot with temperature reconstruction from oxygen isotope of ice. This may provide an independent evidence for the behavior of nitrogen and argon isotopes. In particular, I would like to point out the 9.2ka Event. This event can be clearly seen in Kobashi et al. (2017) reconstructions and oxygen isotope of ice, but not for temperature reconstructions for d15N. This may implicate that fast temperature changes are canceled by firn responses in d15N. Rapid cooling induce fast thickening of firn (increase in d15N) and surface cooling (decrease in d15N), cancelling each other, which possibly cannot be captured by the two firn models with slow firn densification processes. 6. Authors state that the late and early Holocene part of ice has impacted by gas loss. However, the late Holocene ice cores were good quality, and the past 1000 years have the highest quality data available to test the hypotheses (Kobashi et al., 2008). Authors cannot simply state the data is bad quality. Authors must compare the past 1000 years temperature reconstruction for d15N, d40Ar, Kobashi et al (2011), oxygen isotopes, and borehole temperature reconstruction to make your work useful. Note that Kobashi et al. (2011) has higher time resolution than Kobashi et al. (2017).

Detailed comments

Page 1. Line 14. This statement is based on a hypothesis that the firn-models they use are correct. However, the hypothesis is not sufficiently tested to reject another hypothesis that isotopic signals of dNexcess and dNgravi (or isotopically derived firn depth) are real. Therefore, it cannot claim that dNexcess based method is “problematic”. Because the basic assumptions are different between firn model-based method (densification) and dNexcess method, there are no surprises that the Kobashi et al temperature is radically different from Author’s and Buizert temperature reconstructions.

Page 1. Line 18. Early to middle Holocene part of ice is very good quality. Middle to

late Holocene ice is brittle (Kobashi et al., 2008). The authors need to look into physical properties of ice or other information, not only isotopic data to draw conclusions.

P3, line 11. Basic constructs of Goujon and Schwander's firn models are the same. You must explain what is different and what is the same between these two models in the method section. Then, it becomes clear what you are testing with the two models.

Page 4. Line 26, "we argue that only $\delta$15N is suitable as a robust reconstruction target in the high-frequency case". Why only d15N? It is easier to find signals in d15N, but d15Nexcess has also signals (Fig. 1). Also, the resolution and error of the data is different in different time span. For the past 1000 years or around 8200 B.P., time resolution is higher (~10 years) and uncertainty is smaller (more data for each depth).

Page 5. Line 10. The data show opposite of the author's claim. Kobashi et al. (2008) conducted extensive tests for d15N and d40Ar from the same depth range in two different periods. Kobashi et al. (2008) found that d15N and d40Ar/4 from the same depth range have different values in the two periods by about 2 permeg most likely owing to standardization of gasses to atmosphere. The pooled standard deviation of these values are 4 permeg for the both gasses. For the same data, dNexcess has identical number in a permeg scale, indicating that dNexcess is more conservative proxy than d15N or d40Ar because the calculation of dNexcess (d15N-d40Ar/4) cancels out possible gas fractionation during handling or gas loss.

Page 6. The early to middle Holocene ice is in a brittle ice zone, where gases more likely leak from the ice from cracks. In these ice cores, d15N also is affected by gas-loss process (Kobashi et al., 2008). Indeed, d15Nexcess is more robust for gas-ross because the gas ross induces 1:4 for d15N and d40Ar as stated earlier.

Page 6, line 8, The firn depth (dNgrav) can be obtained from the residual of calculation of dNexcess. This could be used to constrain the model. However, we found that if dNexcess are used as inputs, the firn depth cannot be reproduced by the model. You can interpret it as the "firn model" is wrong, or "the data" is wrong.

Page 6. Line 4. “the executed correction” I did not find the explanation on the executed correction on d40Ar for the author’s calculation. I think it may be not enough for the author’s calculation. In Kobashi et al. (2017), we have conducted correction for each segment of time differently, and it worked fine within errors. This is necessary because the ice cores were analyzed in 3 years in different sets of time. In each time span, experimental settings were different, which caused slight shifts in isotope values during the standardization processes.

Page 14. 4.1.1. Absolute temperature is available from borehole temperature reconstructions for GISP2. Why you don’t try to calibrate the temperature to the borehole temperature? As the firn models are not linear, different absolute temperature will create different firn responses. The temperature calculation needs to be calibrated with the borehole temperature data.

Page 15. Line 13. This is the depth where ice core is good condition, and the middle to late Holocene is in the brittle zone. In good ice core zones, gas leaks from ice cores are less important.

Page 16. Line 3. As mentioned earlier, the temperature calculation using dNexcess is based on a different hypothesis. According to the hypothesis, the author’s temperature calculation using the firn model (densification) is not recommended for climate interpretation.

Page 16. Line 4. It is not clear what argon correction the authors used. Argon correction for Kobashi et al. (2015b) and Kobashi et al. (2017) are different. Kobashi et al. (2017) used more advanced correction of different values for different time spans. As mentioned earlier, isotopic experiments introduce slightly different values for both d15N and d40Ar during standardization. Therefore, even without gas loss, ïĄĎT integration method needs to have corrections for ïĄĎT for the different periods of experiments.

Page 17. Line 10. In the Kobashi et al. (2017), we created a method to correct d15Nexcess in different time spans. The correction is minor as most of the variability

in reconstructed temperature are originating in raw dNexcess (Fig S2 in Kobashi et al., 2017).

References:

Kobashi, T., L. Menviel, A. Jeltsch-Thömmes, B. M. Vinther, J. E. Box, R. Muscheler, T. Nakaegawa, P. L. Pfister, M. Döring, M. Leuenberger, H. Wanner, A. Ohmura, Volcanic influence on centennial to millennial Holocene Greenland temperature change, Scientific Reports, 7, Article number: 1441, 2017.

Kobashi, T., J.E. Box, B.M. Vinther, K. Goto-Azuma, T. Blunier, J.W.C. White, T. Nakaegawa, C.S. Andresen, Modern solar maximum forced late twentieth century Greenland cooling, Geophys. Res. Lett., 42, 5992–5999, 2015.

Kobashi, T., K. Kawamura, J. P. Severinghaus, J.-M. Barnola, T. Nakaegawa, B. M. Vinther, S. J. Johnsen., and J. E. Box, High variability of Greenland temperature over the past 4000 years estimated from trapped air in ice core, Geophysical Research Letters, v. 38, L21501, doi:10.1029/2011GL049444, 2011.

Kobashi, T., J. P. Severinghaus, J.-M. Barnola, K. Kawamura, T. Carter, and T. Nakaegawa, Persistent multi-decadal Greenland temperature fluctuation through the last millennium, Climatic Change, 100, 733-756, 2010.

Kobashi, T., J. P. Severinghaus, and K. Kawamura, Argon and nitrogen isotopes of trapped air in the GISP2 ice core during the Holocene epoch (0-11,500 B.P.): Methodology and implications for gas loss processes, Geochimica et Cosmochimica Acta, 72, 4675-4686, 2008.

---

## Referee Comment (RC1) · Anonymous Referee #1 · 7 Dec 2020

Review: Döring, M. and Leuenberger, M. C.: Comparison of Holocene temperature reconstructions based on GISP2 multiple-gas-isotope measurements, Clim. Past Discuss., https://doi.org/10.5194/cp-2020-132, in review, 2020

This manuscript presents the first application of a recently developed algorithm/firn model scheme to the problem of Greenland temperature variability recorded by ice core gas isotope data during the Holocene.

This is my second time reviewing this manuscript (previously as reviewer 2). I commend the authors on taking on board many of my suggestions and carrying out a major restructuring of the paper. I find the manuscript somewhat improved, particularly in

terms of readability. That said, the manuscript is still very long and filled with technical details and close examination of model output that is often deemed highly uncertain. There are a number of long sections that end with conclusion to say that a particular effect is small (e.g. accumulation history) or the authors do not fully understand why their model-based prediction do what they do (e.g. differences in absolute temperature between Schwander and Goujon model as well as the uncertainty in the 15Nexcess fitting between the two models).

After a close reading, a gas isotope specialist will likely be able to gather some useful information. However, I suspect that the paper will be impenetrable for a non-specialist (e.g. a climate modeller interested in examining Holocene temperature variability). Given the specialist nature of Climate of the Past, this is perhaps not a barrier to publication. I mention it here so the authors keep this mind during the response to reviews and it brings up my only major comment.

For a non-specialist, the paper does not provide a clear answer as to which of the various methods will most accurately reconstruct surface temperature. One is even left with the impression that this is perhaps an intractable problem. If this is indeed the overall point of the authors I can understand their hesitancy to push any one particular reconstruction. However, if they see more utility in some more than others, or believe that all should be considered equally, then they should discuss this with the general user in mind. Moreover, they should make their reconstructions available for use, if they believe their results robust. Otherwise the community will continue to use the previously published reconstructions that the paper (sometimes) attempts to argue are not fully robust.

On the other hand, if they believe all their results are not robust then they should state this explicitly. For example, the final sentence of the abstract reads "However, all three reconstruction strategies lead to distinct temperature realizations". What are the implications of this? Should we use just one method or none at all?

Will the author's be making the temperature reconstructions publically available? Similarly, is their code available upon request?

I am surprised not to see any comparison climate model predictions. One way to frame the question of which temperature prediction is realistic would be to compare to model predictions (on centennial to millennial-timescales in Holocene). This would be an entirely different paper, but would show the implications of the different methods. This would also highlight the problem of seasonality changes on d18O.

Specific comments Page 2, line 8. A reference to the original Dansgaard study is appropriate here. The reference to Gierz et al., 2017 is more appropriate if discussing water isotopes during the last interglacial, where even then the literature is quite broad.

Page 2: line 25. The author setup of the paper by suggesting the Buizert approach could be complicated by the d18O "side effects" related to seasonality. Please consider using a different term than "side effect". But moreover, this is an important point, but is never followed up on in the text.

Page 4, line 14. Remove capitalization of "First".

Page 4, line 35. Can you provide a reference for this calculation of the signal smoothing based on resolution? I don't quite follow based on the equation shown why 5.3 is sensible. Are you trying a rough calculation of standard error based on the number of samples within a given window?

Throughout. . .define "cop" in the figures and other parts of the text.

Page 6, line 5. Replace "Non-sufficient" with "insufficient"

Page 6, line 18. A reference to figure our table is needed here as it not "obvious" what the authors a referring to.

Page 6, line 19. Change wording to remove use of "neither". Sentence is complicated because of the double negative

Page 8, lines 11. Please define the phrase "gained data".

Page 14, line 31. Depended should be dependent.

---

## Referee Comment (RC2) · Anonymous Referee #2 · 10 Dec 2020

The paper from Doring et al uses a newly developed algorithm to reconstruct past surface temperatures from nitrogen and argon in ice core, and apply it to previously published GISP2 data.

This version of the paper is much improved in readability compared to the previous submission, but the major flaws in the experimental design are still there, and a lack of climatic interpretation of the results still dramatically limit the value of the work presented here.

My main criticisms are as follow: 1. The authors use a rich dataset, with d15N and d40Ar measurments, but make no use of it, and choose to use only one type of data

at a time. This is regretable, because they are in fact unable to produce a solution that satisfies all of the constraints (see Fig 4 on the right pannels), even though they could, people (Kobashi, for instance) have done it before. Because of this, their reconstructions are no better than what is already published. They do not bring the different existing reconstructions in better agreement. As a result, I do not see the value of the results for non specialists of this particular method. And for specialists, still, they would need a version of the algorithm that can fit all the data at once (which exists and has been published by data producing groups).

2. The authors fail to understand, or explain clearly that, if you fit just d15N, you are basically infering temperature from firn thickness. If you fit d15N excess, you are inferring temperature from the thermal fractionation in the firn. These are almost independent processes, no wonder they produce different answers.

3. When infering temperature from firn thickness (either d15N only or d40Ar only), there are two very important assumptions: 1. firn densification models are perfect, 2. the accumulation scenario is perfect. In this instance of the paper, these two hypotheses have been discussed a bit better, by using two different densification models, and by looking at different accumulation scenarios. When I look at figure 5, I do not conclude that the accumulation scenario does not have any impact, but I al also a bit confused by the fact that the full temperature reconstruction is not shown.

4. An evaluation of the results, not just in comparison to Buizert and Kobashi, but compared to external validation data, like d18O, borehole temperature reconstructions, other sites etc would be needed to demonstrate the value of this work. Here, we are left hanging with inconsistent results that are not fully interpreted.

To sum up, I don't think that there is enough added value in the work presented here compared to what is already published (the data, the inverse method, temperature reconstructions at the same site) to justify publication. I recommend this article to be rejected.

---

## Author Comment (AC1) · 21 Dec 2020

We have attached our replies to the review of Anonymous Referee #1, Anonymous Referee #2 and the short comment of Takuro Kobashi, as a single supplement pdf document. The reply to the review of Anonymous Referee #1 can be found on page 2, the reply to the review of Anonymous Referee #2 on pages 3-4 and the reply to the short comment of Takuro Kobashi is at page 5.

Please also note the supplement to this comment:
https://cp.copernicus.org/preprints/cp-2020-132/cp-2020-132-AC1-supplement.pdf

**Supplement:**

**Comments to the two reviews and the one short comment by Takuro Kobashi:**

We can understand the decision taken by the editor based on the two reviews and the short comment of Takuro Kobashi. At the same time, we would like to mention that we do not agree with most of the comments given by Takuro Kobashi in his short comment as well as Reviewer 2 in this second round, corresponding to Reviewer 1 in the first review round (hereafter referred to as Reviewer 2).

Comment to Reviewer 1:

We are fully aware of the fact that the manuscript is rather difficult to read for a non-specialist. That is why we tried to include as much information as we have gathered during our work on temperature reconstructions using the isotope compositions of nitrogen and argon (and the combined quantity $\delta^{15}N_{excess}$) extracted from ice cores. Obviously, we again failed to transport the complex nature of this reconstruction to the reader. Simply said, and we certainly will take this up in a submission to another journal, it is an issue of signal-to-noise ratio and systematic data uncertainty defining which method should be used for the most robust temperature reconstruction. If we would have access to ideally measured nitrogen and argon isotopes, $\delta^{15}N_{excess}$ would be the best choice. Unfortunately, this is not the case. We hoped to illustrate this issue with Figures 1, 4 and S1 of the manuscript and supplement. Therefore, we have investigated several other possibilities including sole $\delta^{15}N$ or $\delta^{40}Ar$ reconstructions as well as combinations of those.

Comments to Reviewer 2:

Reviewer 2 states (red):

"My main criticisms are as follow: 1. The authors use a rich dataset, with d15N and d40Ar measurements, but make no use of it, and choose to use only one type of data at a time. This is regretable, because they are in fact unable to produce a solution that satisfies all of the constraints (see Fig 4 on the right pannels), even though they could, people (Kobashi, for instance) have done it before"

This is simply wrong; we did investigate the combination of both measurements, i.e. $\delta^{15}N_{excess}$ and made comparisons with Kobashi's (2017) solutions. We must assume that reviewer 2 did not even read the manuscript. In addition, in Kobashi et al. (2017) the remaining misfits for the isotope solutions were not quantified. We recommend studying Figure S1 in the supplement of Kobashi et al. (2017) and judge again.

"2. The authors fail to understand, or explain clearly that, if you fit just d15N, you are basically inferring temperature from firn thickness. If you fit d15N excess, you are inferring temperature from the thermal fractionation in the firn. These are almost independent processes, no wonder they produce different answers."

Here, again we do not know whether Reviewer 2 understands our approach. It is clearly step by step explained in our method paper (Döring and Leuenberger, 2018). We clearly distinguish between firn thickness and temperature influences, the question that we pose in the present manuscript is to which extend are the $\delta^{15}N_{excess}$ usable considering its very low signal-to-noise ratio and systematic offsets toward to too low values (especially in the early and late Holocene). Principally, Reviewer 2 is right when there would not be an issue of uncertainty. However, this is clearly the case as shown for instance by Figures 1, 4 and S1 in the present manuscript.

The reviewer is right by saying: "… no wonder they produce different answers."
Indeed, if you interpret random noise or systematic offsets, it will also result in temperature variations.

3. When inferring temperature from firn thickness (either d15N only or d40Ar only), there are two very important assumptions: 1. firn densification models are perfect, 2. The accumulation scenario is perfect. In this instance of the paper, these two hypotheses have been discussed a bit better, by using two different densification models, and by looking at different accumulation scenarios. When I look at figure 5, I do not conclude that the accumulation scenario does not have any impact, but I al also a bit confused by the fac that the full temperature reconstruction is not shown.

We already were discussing this issue in the first version of the manuscript not only in this second, but indeed we rearranged and incorporated suggestions from you as well as from reviewer 1 on these two issues. We do not state in our manuscript that the accumulation does not have any impact as suggested by reviewer 2! We actually say: "The differences between the accumulation-rates lead to slightly different modelled Δage in the early-Holocene and to a 0.3 K larger cooling for the higher accumulation-rate scenario compared to the two other ones." This sentence refers to Figure 5 of the manuscript.

4. An evaluation of the results, not just in comparison to Buizert and Kobashi, but compared to external validation data, like d18O, borehole temperature reconstructions, other sites etc would be needed to demonstrate the value of this work. Here, we are left hanging with inconsistent results that are not fully interpreted.

It is interesting that Reviewer 2 refers to a comparison of $\delta^{18}O$ of ice. We have to assume that he does not know that Buizert et al., has actually used $\delta^{18}O$ data calibrated to temperature in order to fit the long-term trend of $\delta^{15}N$ to drive their temperature reconstruction. Furthermore, we have compared our temperature reconstructions for $\delta^{15}N$ to the borehole temperature values, see Fig. S5i. A direct comparison between $\delta^{15}N$ derived temperatures and $\delta^{18}O_{ice}$ values have been published in Michael Döring's PhD thesis, see attached Fig. 4.3 below.

To sum up, I don't think that there is enough added value in the work presented here compared to what is already published (the data, the inverse method, temperature reconstructions at the same site) to justify publication. I recommend this article to be rejected.

Reviewer's 2 judgement is based on statements that are simply incorrect when reading our manuscript carefully. It has implications for the temperature reconstructions when using $\delta^{15}N_{excess}$. And, we would like to repeat again that $\delta^{15}N_{excess}$ temperature reconstructions are superior to those using sole $\delta^{15}N$ or $\delta^{40}Ar$ only if the corresponding signal-to-noise ratio is good enough and the causes of the systematic offsets are understood and quantifiable. Otherwise, one had to face significant uncertainties of reconstructed temperature variations but to the point of simply false temperature trends.

Comments to Takuro Kobashi's short comment:

Answer to comment 1:
In Kobashi et al., 2017 it is stated "We circumvented the drifts by allowing slight constant shifts in $\Delta T$ by minimizing the difference between the observed and modeled $\delta 15N$." This points to an adjustment of reconstructed temperatures based on $\delta^{15}N_{excess}$ to modeled $\delta^{15}N$. If this is not applied, reconstructed temperatures deviate considerably from reasonable temperatures as shown in our manuscript, Fig. 4d. If adjustments are made to $\delta^{15}N_{excess}$ evolution, Fig. 4c hybrid method, then $\delta^{15}N$ plays a very important role again. Also "slight" shifts in $\Delta T$ can change the reconstructed temperatures substantially, e.g. when $\Delta T$ changes from negative to positive values over an extended time period, the temporal integration will change from a cooling to a warming trend.

Answer to comment 2:
Indeed, Kobashi et al, 2017 has compared temperature reconstructions to several records. But we also did this, including borehole temperature, $\delta^{18}O$-based (Buizert et al. 2018) as well as Kobashi et al., 2017. Therefore, by comparing it directly with Kobashi et al., 2017, we indirectly also compared it to all other records that has been mentioned by Kobashi et al., 2017. In addition, we have attached Fig. 4.3, i.e. a comparison of $\delta^{15}N$ derived temperatures with Gisp2 $\delta^{18}O_{ice}$ values.

Answer to Comment 3:
Yes, the data will be available.

Answer to comment 4:
Yes, we did see Fig. S5i.

Answer to comment 5:
Yes, we did that see Fig. 6.  In addition, we add here Fig. 4.3 of Michael Döring's thesis i.e. a comparison of $\delta^{15}N$ derived temperatures with Gisp2 $\delta^{18}O_{ice}$ values as well as a comparison to the Renland/Agassiz $\delta^{18}O_{ice}$-based reconstruction of Vinther et al. 2009.

Answer to comment 6:

The late and early Holocene data show the strongest systematic offsets in $\delta^{15}N_{excess}$ (and $\Delta T$) compared to the rest of the record. This issue becomes visible when inverting the data to temperature (Fig. 4d). We do not say that these systematic negative offsets are caused by gas loss, but we stated that the mechanism which causes these offsets works in the same direction (too low $\delta^{15}N_{excess}/\Delta T$ may be caused by to high $\delta^{40}Ar$). This is also visible in supplement figure S3 in Kobashi et al. 2017 (largest offsets between modelled and measured $\delta^{15}N_{excess}/\Delta T$ in these sections).

Figures from PhD Thesis:

T($\delta^{15}$N) vs. GISP2 $\delta^{18}$O$_{ice}$:

[Figure]

Figure 4.3: Comparison of T($\delta^{15}$N) anomaly (blue curves) with $\delta^{18}$O$_{ice}$ (red curves) over the Holocene on different periodic-time bands. (A) Low-pass filtered signals (cut-off-period: 500 yr) display the general trends. (B) Multi-decadal signals (band-pass: 50 yr to 200 yr). (C) Multi-centennial signals (band-pass: 200 yr to 1000 yr). (D) Multi-millennial signals (band-pass: 1 kyr to 4 kyr).

T($\delta^{15}$N) vs. Vinther et al. 2009:

[Figure]

Comparison of GISP2 T($\delta^{15}$N) anomaly with stable-water-isotope ($\delta^{18}$O$_{ice}$) based temperature reconstruction from coastal Greenland sites (*Vinther et al. 2009*).